# Geometric fluctuation of conformal Hilbert spaces and multiple graviton modes in fractional quantum Hall effect

Wang Yuzhu [ORCID][1] & Yang Bo [ORCID][1,2] ✉

Neutral excitations in fractional quantum Hall (FQH) fluids define the incompressibility of topological phases, a species of which can show graviton-like behaviors and are thus called the graviton modes (GMs). Here, we develop the microscopic theory for multiple GMs in FQH fluids and show explicitly that they are associated with the geometric fluctuation of well-defined conformal Hilbert spaces (CHSs), which are hierarchical subspaces within a single Landau level, each with emergent conformal symmetry and continuously parameterized by a unimodular metric. This leads to several statements about the number and the merging/splitting of GMs, which are verified numerically with both model and realistic interactions. We also discuss how the microscopic theory can serve as the basis for the additional Haldane modes in the effective field theory description and their experimental relevance to realistic electron-electron interactions.

Our universe has two important fundamental constants: the speed of light $c$, which parametrizes the Lorentz invariance (from the theory of relativity), and Planck's constant $\hbar$, which parametrizes the quantum fluctuation. It turns out that in a two-dimensional "universe" realized by the quantum Hall effect, we also have two analogous "fundamental constants": The Fermi velocity of the chiral Luttinger liquid of the edge transport is analogous to $c$, while the magnetic length is analogous to $\hbar$. Furthermore, in such a microscopic universe, these two parameters can be tuned experimentally[1]. This leads to rich physics from the interplay of geometry, topology, and emergent symmetry due to strong interactions[2], which can even induce the emergence of the quasiparticles analogous to those theoretically proposed at high energy but have yet been observed in nature. One intriguing example is the gravitons, which are the hypothetical spin-2 bosons from the quantization of gravitational field[3,4]. There also exist the analogous graviton modes (GMs) in a fractional quantum Hall (FQH) fluid[5-7] is a two-dimensional quantum fluid of electrons subject to a strong magnetic field at low temperatures. These modes are the quadrupole gapped excitations of the quantum Hall effect that emerge from the geometric fluctuation of the topological ground state, encoding topological information about their respective FQH phases. Their dynamics leads to rich physics ranging from ground state incompressibility to the dynamical phase transitions of the low-lying excitations[8,9].

The effective field theory studying these modes has been proposed using the Newton-Cartan formalism, and various experimental proposals for the observation of these modes have been put forward[10-19]. The standard technique in probing neutral excitations is to use the inelastic photon scattering[20-24]. The coupling between the GMs and the acoustic waves can also be used to simulate the behavior of gravitons interacting with the gravitational waves[25]. Meanwhile, the microscopic theory of the GMs with model Hamiltonians has also been established to provide insights for the experiments, where model Hamiltonians for the GMs have been constructed for FQH fluids at different filling factors[9,26]. Recently numerical results have implied the signature of multiple GMs in FQH states, the microscopic understanding of which is under development[27,28]. Given that most of the research on GMs is based on effective field theories and numerical analysis with model wavefunctions[28], a detailed microscopic theory is needed for a complete characterization of the emergence and interaction between different GMs.

[1]School of Physical and Mathematical Sciences, Nanyang Technological University, Singapore 639798, Singapore. [2]Institute of High Performance Computing, A*STAR, Singapore 138632, Singapore. ✉e-mail: yang.bo@ntu.edu.sg

In this work, we show analytically that multiple GMs are a generic feature of FQH fluids, from the splitting of the long-wavelength limit of the Girvin-MacDonald-Platzman (GMP) mode[5] in different subspaces in a single Landau level (LL). Using the analytic tools we developed earlier[9], we demonstrate that the number of observable GMs is dynamical in nature and is only meaningful when referring to specific interaction Hamiltonians. Each GM can be interpreted as the metric fluctuation of a conformal Hilbert space (or the null spaces of model Hamiltonians, as explained later) within a single LL. For short-range two-body interactions, we show all non-Laughlin FQH states around the filling factor $\nu = 1/(2n)$ with $n > 1$, including the interacting composite fermion (CF) states, have at least two GMs. In particular, the Jain states at $\nu = N/(2nN \pm 1)$ and the Pfaffian states at $\nu = 1/(2n)$ all have two GMs if $n, N > 1$. The Laughlin states ($N = 1$) and the Jain states with $n = 1$ all have a single GM. This agrees with the special cases studied numerically in both refs. [27],[28] at $\nu = 2/7, 2/9, 1/4$, at the same time providing an analytic explanation and geometric interpretation to their numerical observations. Furthermore, the microscopic theory can easily predict the chirality of the gravitons[19] without numerical computations.

This work is organized as follows: First we conceptually introduce the geometrical origin of the GMs and the hierarchical structure of the conformal Hilbert spaces (CHSs) as the null spaces of model Hamiltonians, the combination of which leads to the microscopic explanation of the emergence of multiple GMs, with the spectral function calculated from the single-mode approximation wave function to distinguish these GMs in the Hilbert space. Then we focus on the GMs in the CHSs defined by short-range two-body interactions and non-Abelian three-body interactions, where both analytical and numerical evidence shows the signature of multiple GMs; This adds yet another tool for the experimental probing of topological orders in low-temperature, two-dimensional electronic systems where the Coulomb interaction can be slightly tuned and how such orders are affected by the conformal symmetry that may or may not be fully realized in experiments; How our theory serves as the basis for the effective field theory is explained with more technical details, in the discussion section, where we show the necessity of additional Haldane modes depends on the proper identification of the base space of such theories.

## Results

### The cyclotron and guiding center metric

It is useful to consider the simple case of the integer quantum Hall effect (IQHE), which are topological phases from fully filled LLs. We are dealing with the following full Hamiltonian:

$$\hat{H} = \sum_{i=1}^{N_e} \frac{1}{2m} \tilde{g}^{ab} \hat{\pi}_{ia} \hat{\pi}_{ib} + \hat{V}_{\text{int}} \qquad (1)$$

Here, $\hat{\pi}_{ia} = \hat{p}_{ia} + eA_{ia}$ denotes the dynamical momentum operator of the $i$-th electron ($\hat{p}_{ia}$ is the canonical momentum and $A_i$ is the external vector potential), with the commutation rules $[\hat{\pi}_{ia}, \hat{\pi}_{jb}] = i\delta_{ij}\epsilon_{ab}/\ell_B^2$. The magnetic field is $B = \epsilon^{ab}\partial_a A_{ib}$ and the magnetic length is $\ell_B = \sqrt{1/eB}$. We assume the cyclotron energy is the dominant energy scale, so any LL mixing induced by electron-electron interaction can be perturbatively captured by few-body interaction of the second term $\hat{V}_{\text{int}}$, which now describes the dynamics only within a single LL[29–33]. The important note here is that the Hilbert space of a single LL, which we will refer to as the lowest LL (LLL) without loss of generality, is parametrized by the unimodular metric $\tilde{g}^{ab}$ in Eq. (1), which is physically the effective mass tensor. Quantum fluctuations around this metric thus lead to density modes in higher LLs, which we can term "cyclotron gravitons". The energy of this GM is very high, with a large magnetic field. It is the only GM for the IQHE since the LLL is fully filled.

**Table 1 | Definition of various symbols used in the text**

| | |
|---|---|
| $\hat{R}_i^a$ | Guiding center operator |
| $\hat{\bar{\rho}}_{\mathbf{q}}$ | Guiding center density operator |
| $\delta\bar{\rho}_{\mathbf{q}}$ | Regularized guiding center density operator |
| $\hat{\pi}_{ia}$ | Dynamical momentum operator |
| $\bar{g}^{ab}$ | Guiding center metric |
| $\tilde{g}^{ab}$ | Cyclotron metric |
| $g_\alpha^{ab}$ | Guiding center metric of CHS $\mathcal{H}_\alpha$ |
| $S_{\mathbf{q}}$ | Regularized guiding center structure factor |
| $\hat{V}_\alpha^{k\text{bdy}}$ | $k$-body pseudopotential |
| $\hat{\mathcal{V}}_\alpha^{k\text{bdy}}$ | Model Hamiltonian defined by $\sum_{i=1}^\alpha \lambda_i \hat{V}_i^{k\text{bdy}}$ |
| $\mathcal{H}_\alpha^{k\text{bdy}}$ | CHS determined by $\hat{\mathcal{V}}_\alpha^{k\text{bdy}}$ |

Note that due to fermionic statistics, the constant coefficients $\lambda_i$ in $\hat{\mathcal{V}}_\alpha^{k\text{bdy}}$ might vanish. For example, $\hat{\mathcal{V}}_3^{2\text{bdy}} = \lambda_1 \hat{V}_1^{k\text{bdy}} + \lambda_3 \hat{V}_3^{k\text{bdy}}$ with $\lambda_2 = 0$.

For FQHE in a partially filled LL, the dynamics is determined entirely by the guiding center coordinates $\bar{R}^a = \hat{r}^a - \epsilon^{ab}\hat{\pi}_b \ell_B^2$ with the commutation rules $[\bar{R}^a, \bar{R}^b] = i\ell_B^2 \epsilon^{ab}, [\bar{R}^a, \tilde{R}^b] = -i\ell_B^2 \epsilon^{ab}, [\tilde{R}^a, \tilde{R}^b] = 0$, where we define $\tilde{R}^a = \ell_B^2 \epsilon^{ab}\hat{\pi}_b$ as shown in Table 1). This implies the interaction energy $\hat{V}_{\text{int}}$ is a functional of $\bar{R}_i$ only and commutes with the kinetic energy. It can be explicitly expressed as:

$$\hat{V}_{\text{int}} = \int d^2\mathbf{q} V_{|\mathbf{q}|} \bar{\rho}_{\mathbf{q}} \bar{\rho}_{-\mathbf{q}} \qquad (2)$$

where $\bar{\rho}_{\mathbf{q}} = \sum_i e^{i\mathbf{q}\cdot\bar{\mathbf{R}}_i}$ is the guiding center density operator. For rotationally invariant systems, we have a new unimodular metric $\bar{g}^{ab}$ defining distance in the momentum space $|\mathbf{q}| = \sqrt{\bar{g}^{ab}q_a q_b}$, which is physically independent of $\tilde{g}^{ab}$. We illustrate a complete analogy to the "cyclotron graviton" in the IQHE by using the simple example of $\hat{V}_{\text{int}} = \hat{V}_1^{2\text{bdy}}$, or the model Hamiltonian for the Laughlin $\nu = 1/3$ state (Commonly-used notations of model Hamiltonians are listed in Table 2). Just like the LLL, which is the null space of the kinetic energy parameterized by $\tilde{g}^{ab}$, the null space of $\hat{V}_1^{2\text{bdy}}$ (spanned by the Laughlin ground state and quasiholes) is parametrized by $\bar{g}^{ab}$. The quantum fluctuation around $\tilde{g}^{ab}$ gives the "cyclotron graviton" outside of LLL, while that of $\bar{g}^{ab}$ gives the well-known graviton or quadrupole mode (the long-wavelength limit of the GMP mode) outside of the $\hat{V}_1^{2\text{bdy}}$ null space[7].

We want to emphasize that the arguments above apply to any $\hat{V}_{\text{int}}$ with an incompressible ground state. Thus, generally speaking, all FQH states have at least two GMs due to the structure of the full Hamiltonian: the cyclotron GM residing in higher LLs (which has very high energy due to the large magnetic field) and at least one guiding center GM within the LLL. Although for the rest of this work, we will ignore the cyclotron GMs and focus on the dynamics within the LLL, we can use the cyclotron GMs as examples to understand the emergence of multiple guiding center GMs from $\hat{V}_{\text{int}}$.

### The hierarchy of conformal Hilbert spaces

Given the rich algebraic structure of the Hilbert spaces like the LLL and the $\hat{V}_1^{2\text{bdy}}$ null space, we term them as conformal Hilbert spaces (CHSs) because they are believed to be generated by the conformal operators (i.e., the Virasoro algebra)[34,35]. In many cases, such Hilbert spaces are spanned by degenerate (zero energy) many-body states of special local Hamiltonians, including the well-known generalized pseudopotentials, that physically project into the angular momentum sectors of a cluster

of electrons[36,37]. The null spaces of these Hamiltonians have conformal symmetry in the thermodynamic limit. Those zero energy states are the ground states and quasiholes of a particular FQH phase (though some are believed to be gapless phases)[38,39]. More importantly, like the Hilbert space of the LLL (or any other single LL), such CHSs are built up with quasiparticles, which are emergent particles from LL projection and strong interaction. In the LLL, the quasiparticles are simply electrons projected into a single LL, while in other CHSs, they can be abelian or non-abelian anyons[40–46].

Let $\mathcal{H}_\alpha$ be one of these CHSs, and the null space of the corresponding model Hamiltonian $\hat{V}_\alpha$. Just like in Eq. (2), $\hat{V}_\alpha$ contains a

**Table 2 | The CHSs are defined as the null spaces of the corresponding model Hamiltonians**

| CHS | Model Hamiltonian |
| --- | --- |
| $\mathcal{H}_{\text{Fibonacci}}$ | $\hat{V}_6^{\text{4bdy}}$ |
| $\mathcal{H}_{\text{MR}}$ | $\hat{V}_3^{\text{3bdy}}$ |
| $\mathcal{H}_{\text{Gaffnian}}$ | $\lambda_1\hat{V}_3^{\text{3bdy}} + \lambda_2\hat{V}_5^{\text{3bdy}}$ |
| $\mathcal{H}_{\text{Haffnian}}$ | $\lambda_1\hat{V}_3^{\text{3bdy}} + \lambda_2\hat{V}_5^{\text{3bdy}} + \lambda_3\hat{V}_6^{\text{3bdy}}$ |
| $\mathcal{H}_{\text{Laughlin}-1/3}$ | $\hat{V}_1^{\text{2bdy}}$ |
| $\mathcal{H}_{\text{Laughlin}-1/5}$ | $\lambda_1\hat{V}_1^{\text{2bdy}} + \lambda_2\hat{V}_3^{\text{2bdy}}$ |

Here, $\lambda_i$ can be any constant coefficient.

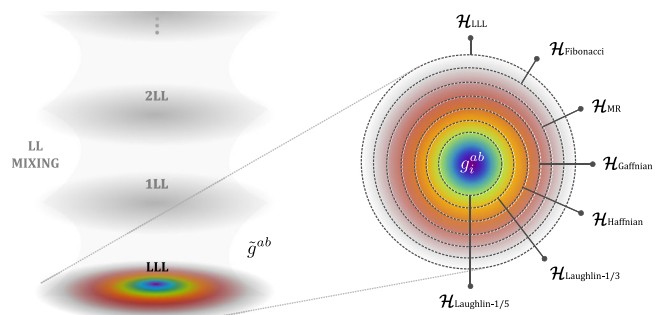

**Fig. 1 | Intrinsic metrics in FQH states.** The left panel shows that the fluctuations of the cyclotron metric $\tilde{g}_{ab}$ originate from the LL mixing. The right panel shows the hierarchical structure of the CHSs (denoted by dashed circles with different colors) of the corresponding model Hamiltonians (more details can be found in Table 2) in the lowest Landau level. Each of these spaces can have its own metric $g_i^{ab}$, the fluctuation around which can potentially lead to multiple GMs in a single Landau level.

guiding center metric $g_\alpha^{ab}$, and $\mathcal{H}_\alpha$ continuously depends on it. This is the geometric aspect we would like to introduce to the CHSs, and each of them can be completely characterized by a triplet of $\{\mathcal{H}_\alpha, \hat{V}_\alpha, g_\alpha^{ab}\}$. For the special case where $\mathcal{H}_\alpha = \mathcal{H}_{\text{LLL}}$, the entire Hilbert space of the LLL, $\hat{V}_\alpha$ is the kinetic energy Hamiltonian and $g_\alpha^{ab} = \tilde{g}^{ab}$ is the cyclotron metric or the effective mass tensor. All other CHSs are subspaces of $\mathcal{H}_{\text{LLL}}$.

In Fig. 1, we illustrate a hierarchical structure of different $\mathcal{H}_\alpha$ in the LLL[9]. For a given $\mathcal{H}_\alpha$, it is possible to find another $\mathcal{H}_\beta \subset \mathcal{H}_\alpha$. If we fix $g_\alpha^{ab}$, we can still define a $\mathcal{H}_\beta$ freely parametrized by $g_\beta^{ab}$ that is entirely within $\mathcal{H}_\alpha$. Since the cyclotron coordinates and guiding center coordinates commute, this is straightforward for $\mathcal{H}_\alpha = \mathcal{H}_{\text{LLL}}$. For other pairs of CHSs, such geometric tuning can only be realized with the following Hamiltonian:

$$\hat{V}_{\text{int}} = \lambda_\alpha \hat{V}_\alpha + \lambda_\beta \hat{V}_\beta \tag{3}$$

with $\lambda_\alpha \gg \lambda_\beta > 0$ (the metric dependence of $\hat{V}_{\alpha,\beta}$ is implicit). For any ground state $|\psi_0\rangle \subset \mathcal{H}_\beta$ of Eq. (3) we can thus define two types of area-preserving deformation:

$$|\psi_1^\chi\rangle \sim \lim_{|\chi|\to 0} \hat{P}_\alpha \hat{U}(\chi)|\psi_0\rangle \sim \lim_{|\mathbf{q}|\to 0} \hat{P}_\alpha \delta\bar{\rho}_\mathbf{q}|\psi_0\rangle \tag{4}$$

$$|\psi_2^\chi\rangle = \left(\hat{\mathbb{I}} - \hat{P}_\alpha \hat{U}(\chi)\right)|\psi_0\rangle \tag{5}$$

where $\hat{U}(\chi) = e^{i\chi_{ab}\hat{\Lambda}^{ab}}$ is the unitary operator inducing the squeezing and rotation of the guiding center metric[7,47], with $\hat{\Lambda}^{ab} = \frac{1}{4\ell^2}\sum_i\{\bar{R}_i^a, \bar{R}_i^b\}$, and the determinant of the symmetric tensor $|\chi|$ parametrize the squeezing; $\delta\bar{\rho}_\mathbf{q} = \bar{\rho}_\mathbf{q} - \langle\psi_0|\bar{\rho}_\mathbf{q}|\psi_0\rangle$ is the regularized guiding center density operator, and Eq. (4) has been established in ref. 6. Here, $\hat{P}_\alpha$ is the projection into $\mathcal{H}_\alpha$ so that $\hat{V}_\alpha|\psi_1^\chi\rangle = 0$, and $|\psi_1^\chi\rangle$ is associated with the geometric deformation of $\mathcal{H}_\beta$. Eq. (5) is entirely outside of $\mathcal{H}_\alpha$, and in some cases, it will vanish, as we will see later. If it is non-vanishing, then $|\psi_2^\chi\rangle$ is associated with the geometric deformation of $\mathcal{H}_\alpha$. This geometric description forms the basis of possible multiple GMs in different FQH phases, dictated by "model Hamiltonians" in the form of Eq. (3). It can be resolved by realistic Hamiltonians close to those model Hamiltonians.

## Emergence of multiple GMs

Within this framework, let us start with a collection of CHSs, $\{\mathcal{H}_k, \hat{V}_k, g_k^{ab}\}$. We will ignore the cyclotron GM, so all these are subspaces of the LLL, and also with a hierarchical structure $\mathcal{H}_{k+1} \subset \mathcal{H}_k$.

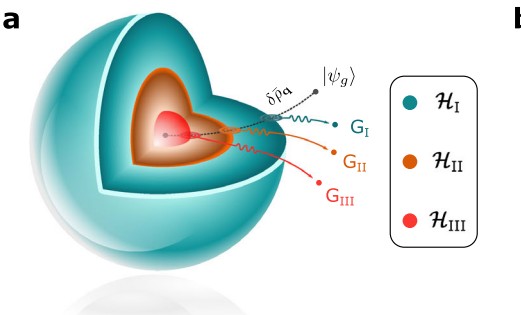
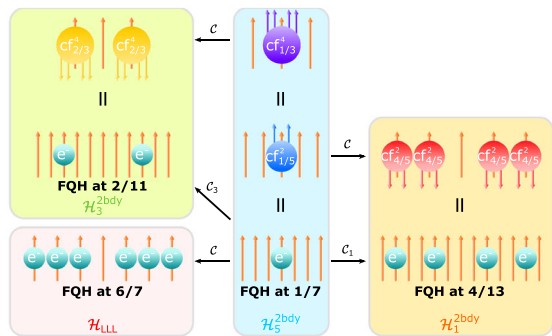

**Fig. 2 | Multiple graviton modes and composite fermionization. a** An illustration of the hierarchical structure of three CHSs and the ground state $|\psi_0\rangle$ within $\mathcal{H}_{\text{III}}$ (red sphere) in the Hilbert space. The corresponding GMP mode $|\psi_g\rangle$ is outside $\mathcal{H}_1$ so one can imagine the regularized guiding center density operator acting on the ground state goes through three CHSs, leading to three emergent GMs because of the fluctuation around the metric of each of the CHSs. **b** PH conjugate of Laughlin

states within different CHSs. Here, $\mathcal{C}_i$ denotes the PH conjugate within $\mathcal{H}_i^{\text{2bdy}}$, and $\mathcal{C}$ denotes the PH conjugate within a single LL or a single CF level. Arrows represent magnetic fluxes, and the CFs denoted by $\text{cf}_{\nu'}^n$, consisted of one electron and $n$ fluxes, form a CF FQH state at $\nu'$. Note that the red ($\text{cf}_{4/5}$) and the yellow ($\text{cf}_{2/3}$) CFs are anti-CFs with the fluxes opposite to the external field.

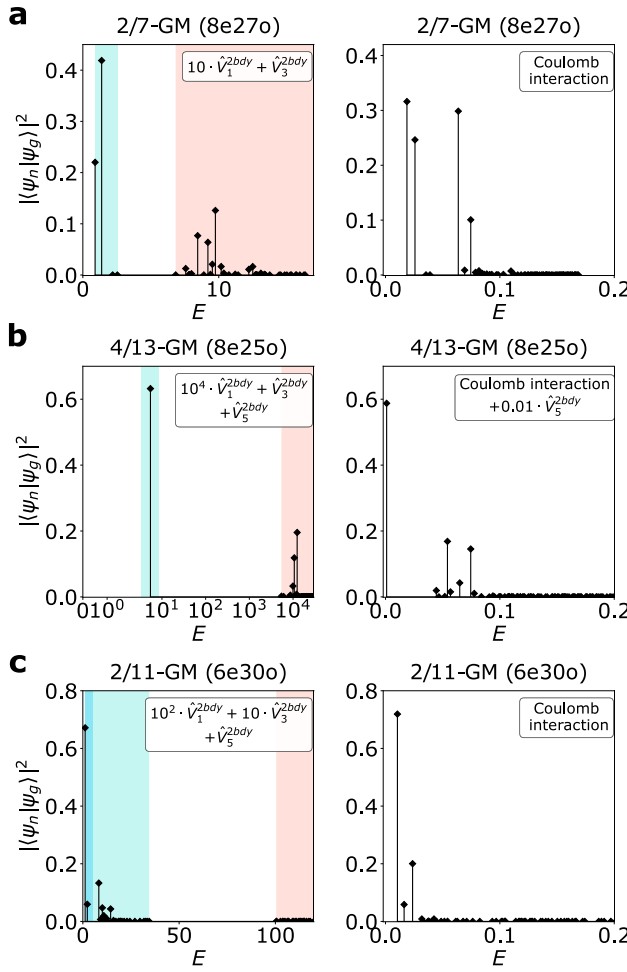

**Fig. 3 | Spectral functions from exact diagonalization. a–c** Graviton peaks of the FQH states with the filling factor $\nu = 2/7$, 4/13 and 2/11. The GM of the FQH state with filling factor $\nu$ is called $\nu$-GM for simplicity, followed by the system size, i.e., $N_e$ electrons and $N_o$ orbitals denoted by $N_e$e$N_o$o. The blue, the turquoise, and the red region denote $\mathcal{H}_3^{2bdy}$, $\mathcal{H}_1^{2bdy}$ and the complement of $\mathcal{H}_1^{2bdy}$ correspondingly, from which one can clearly see the gaps between different sectors. A graviton peak can be distinguished by a bunch of adjacent non-zero stems within a specific sub-Hilbert space, corresponding to a scattering peak that can be observed in experiments. For the FQH states with $\nu = 2/7$ and 4/13 in **a** and **b**, model Hamiltonians show similar signatures of two peaks in the spectral functions as Coulomb interactions (a small $\hat{V}_5^{2bdy}$ is added to the Coulomb interaction in **b** for stabilizing the proper ground state). The corresponding overlap between the model state and the ground state of the Coulomb interaction can be found in Table 3. Source data are provided as a Source Data file.

The model Hamiltonian for understanding the GMs is given by:

$$\hat{V}_{int} = \sum_{k=1}^{m} \lambda_k \hat{V}_k, \qquad \lambda_k \gg \lambda_{k+1} \qquad (6)$$

All metrics $g_k^{ab}$ in the Hamiltonian are arbitrary, and without loss of generality, we can set them as $g_k^{ab} = \mathbb{I}_2$, since the GMs are quantum fluctuations around these fixed metrics. Let the ground state of Eq. (6) be $|\psi_0\rangle \in \mathcal{H}_m$, so the quadrupole excitation is obtained from the long-wavelength limit of the GMP mode or single-mode approximation defined as follows[5,6]:

$$|\psi_g\rangle = \lim_{\mathbf{q} \to \mathbf{0}} \frac{1}{\sqrt{S_{\mathbf{q}}}} \delta\bar{\rho}_{\mathbf{q}} |\psi_0\rangle \qquad (7)$$

Note that $|\psi_g\rangle$ and $|\psi_0\rangle$ are orthogonal, and $S_{\mathbf{q}}$ is the regularized guiding center structure factor with $\lim_{\mathbf{q} \to \mathbf{0}} S_{\mathbf{q}} \sim \eta_s |\mathbf{q}|^4$, which is fully determined by $|\psi_0\rangle$. The Haldane bound dictates that the value of $\eta_s$ gives the upper bound to the topological shift of $|\psi_0\rangle$[47].

The important question here is which CHS does $|\psi_g\rangle$ reside. If $|\psi_g\rangle \in \mathcal{H}_k$ and $|\psi_g\rangle \notin \mathcal{H}_{k+1}$, then obviously there is no GM associated with the quantum fluctuation around $g_k^{ab}$ since such fluctuation will bring us out of $\mathcal{H}_k$. However, $|\psi_g\rangle$ can be decomposed into multiple modes, each within $\mathcal{H}_{k' > k}$ but outside of the $\mathcal{H}_{k'+1}$, associated with the quantum fluctuation around $g_{k'+1}^{ab}$, as long as $|\psi_0\rangle \in \mathcal{H}_{k'+1}$. This is most easily seen by computing the spectral function defined below:

$$I(E) = \sum_n |\langle \psi_n | \psi_g \rangle|^2 \delta(E - E_n) \qquad (8)$$

where $|\psi_n\rangle$, $E_n$ are eigenstates and eigenenergies of Eq. (6). Given that $\lambda_k \gg \lambda_{k+1}$, we will see $m - k$ distinct peaks well-separated in energy, corresponding to $m - k$ GMs, each with transparent geometric interpretation as illustrated in Fig. 2a. The spectral sum rule for the guiding center structure factor will be satisfied from all the contributions of these GMs, as long as $|\psi_g\rangle$ lives completely within $\mathcal{H}_k$.

A number of analytical results have been derived in our previous works, which are rigorous in the thermodynamic limit and useful in determining which CHS $|\psi_g\rangle$ resides in[9]. Let us first take $\hat{V}_{int}$ in Eq. (6) as a sum of short-range two-body interactions as follows:

$$\hat{V}_{int} = \sum_{i=1}^{n} \lambda_i \hat{V}_{2i-1}^{2bdy} \qquad (9)$$

with $\hat{V}_{2i-1}^{2bdy}$ as the $(2i-1)^{th}$ Haldane pseudopotentials (PPs), where fermionic statistics has been considered. Thus, the corresponding null spaces $\mathcal{H}_n^{2bdy}$ is spanned by the Laughlin ground state and quasiholes at $\nu = 1/(2n+1)$. Here, we can prove analytically[9] that $|\psi_g\rangle$ of the Laughlin phase at $\nu = 1/(2n+1)$ resides within $\mathcal{H}_{n-1}^{2bdy}$ but completely outside of $\mathcal{H}_n^{2bdy}$ (we take $\mathcal{H}_0^{2bdy} = \mathcal{H}_{LLL}$), which is saturated by the ground state and quasiholes. Thus, there can only be one GM and one peak in the spectral function associated with the metric fluctuation of $g_n^{ab}$. There are, however, many other FQH states that are incompressible with Eq. (6) but not in $\mathcal{H}_n^{2bdy}$. These include the Jain states at $\nu = N/(2nN+1)$, $N > 1$ and their PH conjugate states (within $\mathcal{H}_{n-1}^{2bdy}$) at $\nu = N/(2nN-1)$, $N > 1$. Note that all these states still resides within $\mathcal{H}_{n-1}^{2bdy}$, though the ground state and quasiholes do not saturate $\mathcal{H}_{n-1}^{2bdy}$. One can prove analytically that their corresponding $|\psi_g\rangle$ all satisfies $|\psi_g\rangle \in \mathcal{H}_{n-2}^{2bdy}$. While they are again completely outside of $\mathcal{H}_n^{2bdy}$, now they have spectral weights within $\mathcal{H}_{n-1}^{2bdy}$. Thus, each of those states will have two GMs with respect to Eq. (6), a generic result agreeing with some special cases studied before[27,28]. These two GMs are associated with the fluctuation of the metric $g_n^{ab}, g_{n-1}^{ab}$, and they can also be understood via clustering properties in parton constructions[28].

Non-abelian FQH states are fascinating in strongly correlated topological systems. Their GMs can also be predicted similarly, assuming that they can be stabilized by realistic interactions adiabatically connected to Eq. (6). For example, the Pfaffian state at $\nu = 1/(2n)$ can be understood as a condensate of paired composite fermions, each with one electron attached to $2n$ magnetic fluxes[48,49]. The model states of these FQH phases all live within $\mathcal{H}_{n-1}^{2bdy}$. For short-range two-body interactions, they will also all have two GMs, one within $\mathcal{H}_{n-1}^{2bdy}$, and the other within $\mathcal{H}_{n-2}^{2bdy}$.

It is also interesting to note that with short-range two-body interaction, all FQH states related to the Laughlin states by particle-hole (PH) conjugation will have at most two GMs. There can be multiple well-defined PH conjugations for each Laughlin state in different Laughlin CHSs, not just within LLL (where PH conjugation relates the state at $\nu$ to $1 - \nu$). This is because the Laughlin state at $\nu = 1/(2n + 1)$ can also be reinterpreted as a Laughlin state of CFs with each electron

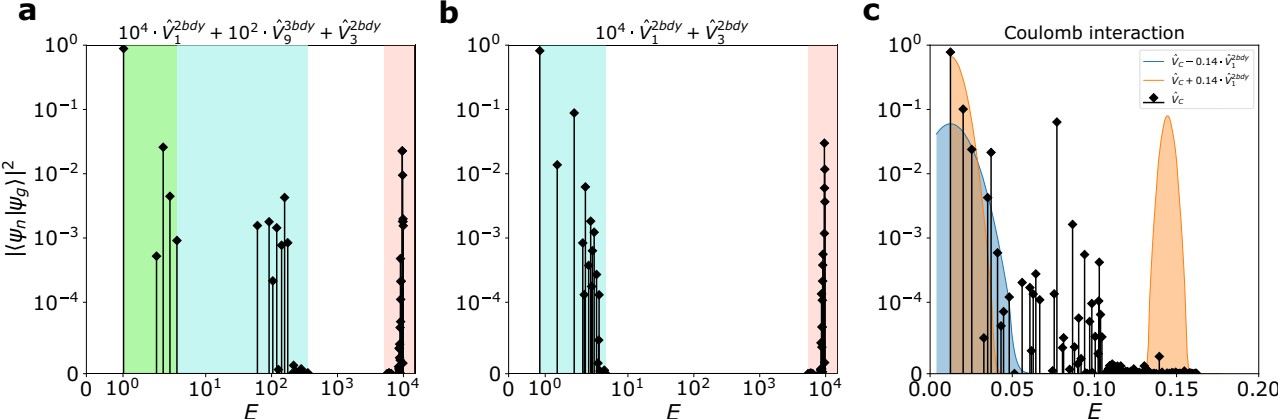

**Fig. 4 | Spectral functions of the FQH state at $v = 2/9$ with 8 electrons. a** Model Hamiltonian. Two peaks within $\mathcal{H}_1^{2\text{bdy}}$ can be seen in the spectral function, where the green, the turquoise, and the red sector denotes the null space of $\hat{V}_9^{3\text{bdy}}$, $\mathcal{H}_1^{2\text{bdy}}$ and the complement of $\mathcal{H}_1^{2\text{bdy}}$. **b** Model Hamiltonian without $\hat{V}_9^{3\text{bdy}}$. The two peaks within $\mathcal{H}_1^{2\text{bdy}}$ merge after $\hat{V}_9^{3\text{bdy}}$ is removed from the Hamiltonian. These graviton modes have the same chirality because the FQH state at $v = 2/9$ has no PH conjugation in any of the CHSs. A slightly exaggerated ratio between different PPs is adopted to clearly show the signature of different CHSs. **c** Coulomb interaction ($\hat{V}_C$) with modifications. Two peaks can also be clearly observed. In experiments, one can tune $\hat{V}_1^{2\text{bdy}}$ to increase (orange area) or decrease (blue area) the separation to get peaks resolved better. Note that for clarity, the peaks with modified $\hat{V}_1^{2\text{bdy}}$ have been smoothed by Gaussians, with the mean and the maximum given by the highest stem, and the standard deviation approximately determined by the stem at the boundary of the peak. Source data are provided as a Source Data file.

attached to $2k$ magnetic fluxes ($k < n$) at the CF fractional filling factor $v^* = 1/(2(n - k) + 1)$[50,51] as Fig. 2b shows. For example, the Laughlin 1/7 state of electrons can be reinterpreted as a $v = 1/5$ state of $cf^2$(one electron bound with two fluxes) within $\mathcal{H}_1^{2\text{bdy}}$. We can then take the PH conjugation within $\mathcal{H}_1^{2\text{bdy}}$, giving the $v = 4/5$ state of $cf^2$, corresponding to the $v = 4/13$ Jain state (of the electron filling, in CF theory it is an interacting CF state). Another example is the FQH state at $v = 2/7$ (a $v = 2/3$ state of $cf^2$), which is the PH conjugate of the Laughlin-1/5 state (a $v = 1/3$ state of $cf^2$) within $\mathcal{H}_1^{2\text{bdy}}$. Due to PH conjugation, we know immediately that there are two GMs with opposite chirality for these two states, as shown in Fig. 3. FQH states with one PH conjugation (e.g., $v = 2/9$) will give all GMs of the same chirality. The chirality of the GMs can thus be predicted without involving numerical calculations. It is also consistent with Ref. 52, since with PH conjugation, the corresponding FQH ground state will not be annihilated by any local Hamiltonians.

We can thus rigorously define the PH conjugate of CFs within $\mathcal{H}_k^{2\text{bdy}}$, at the CF filling factor of $v^* = 2(n - k)/(2(n - k) + 1)$, corresponding to the interacting CF states at electron filling factor $v = 2(n - k)/(2(n - k)(2k + 1) + 1)$. Each Laughlin state at $v = 1/(2n + 1)$ thus have $n$ PH conjugate state with $k = 0, 2, \cdots n - 1$, with $k = 0$ the usual PH conjugate state in LLL at $v = 2n/(2n + 1)$. Since the $v = 2(n - k)/(2(n - k)(2k + 1) + 1)$ state lives entirely within $\mathcal{H}_k^{2\text{bdy}}$ and entirely outside of $\mathcal{H}_k^{2\text{bdy}}$, one can rigorously show[9] its GM lives entirely within $\mathcal{H}_{k-1}^{2\text{bdy}}$ for $k > 0$, leading to two GMs. These two GMs come from the fluctuation of $g_k^{ab}$, as well as the fluctuation of the metric defining the CHS of $v = 2(n - k)/(2(n - k)(2k + 1) + 1)$. Since we are taking the PH conjugate within $\mathcal{H}_k^{2\text{bdy}}$, the GM within $\mathcal{H}_k^{2\text{bdy}}$ will also have the opposite chirality as the one outside of it. For $k = 0$, the anti-Laughlin state at $v = 2n/(2n + 1)$ only has one GM since there is no additional CHS defined by the two-body interaction within LLL that also contains the CHS of $v = 2n/(2n + 1)$.

We now illustrate that the number of GMs is a dynamic property strongly dependent on the interaction. Let us first look at the Jain state at $v = 2/9$, corresponding to the $v^* = 2$ state of the CFs with each electron bound to four magnetic fluxes, or the $v^* = 2/5$ state of the CFs with each electron bound to two magnetic fluxes. We have argued before that with short-range two-body interactions; this FQH state has two GMs. It is important to note, however, while the CHS of this FQH state is well-defined from the CF construction, such construction does not allow an exact model Hamiltonian within the LLL. The two-body interaction only defines its CHS approximately, though to a

very good level of accuracy. A better microscopic Hamiltonian is given as follows:

$$\hat{V}_{\text{int}} = \hat{\mathcal{V}}_n^{3\text{bdy}} = \sum_{i=3}^{n} \hat{V}_i^{3\text{bdy}} \tag{10}$$

where $\hat{V}_i^{3\text{bdy}}$ are the three-body PPs[37]. Note there is no $\hat{V}_4^{3\text{bdy}}$ due to fermionic statistics, and $\hat{V}_9^{2\text{bdy}}$, $\hat{V}_{11}^{3\text{bdy}}$ are doubly degenerate, so here we take them as an arbitrary linear combination (the CHS is invariant). The unique highest density ground state of Eq. (10) with $n = 11$ has a very high overlap with the Jain $v = 2/9$ state (-0.99 for eight electrons). While its quasihole counting is non-Abelian, one could conjecture that the ground state is topologically equivalent to the Jain state, in analogy to the Gaffnian state and the Jain $v = 2/5$ state that has been studied before[35,53].

While important by themselves, such subtleties do not really affect our discussions about GMs, which are gapped excitations. The main message here is that the null spaces of $\hat{\mathcal{V}}_n^{3\text{bdy}}$ give a family of CHSs beyond the Laughlin CHSs discussed before. The Moore-Read, Gaffnian, and Haffnian CHSs are illustrated in Fig. 1, corresponding to the case of $n = 3, 5, 6$, respectively. Let the null space of $\hat{\mathcal{V}}_n^{3\text{bdy}}$ be $\mathcal{H}_n^{3\text{bdy}}$, and it is easy to check that $\mathcal{H}_{11}^{3\text{bdy}} \subset \mathcal{H}_9^{3\text{bdy}} \subset \mathcal{H}_1^{2\text{bdy}}$. Note that the ground state of $v = 2/9$ resides in $\mathcal{H}_{11}^{3\text{bdy}}$, and it has very high overlap with the ground state of $\hat{V}_3^{2\text{bdy}}$. We can thus construct the following Hamiltonian:

$$\hat{V}_{\text{int}} = \lambda_1 \hat{V}_1^{2\text{bdy}} + \lambda_2 \hat{\mathcal{V}}_9^{3\text{bdy}} + \hat{V}_3^{2\text{bdy}} \tag{11}$$

with $\lambda_1 \gg \lambda_2 \gg 1$. In addition to the original two GMs (one from the metric fluctuation of $\mathcal{H}_{11}^{3\text{bdy}}$, or the CHS of the $v = 2/9$ phase, and the other from the metric fluctuation of $\mathcal{H}_1^{2\text{bdy}}$), there will be a third GM from the metric fluctuation of $\mathcal{H}_9^{3\text{bdy}}$, easily observable from the spectral function. Suppose we tune $\lambda_2$ to zero. In that case, the two peaks corresponding to the GMs of the lower energies will gradually merge to become a single peak within $\mathcal{H}_1^{2\text{bdy}}$, accounting only for the metric fluctuation of the CHS of the $v = 2/9$ as shown in Fig. 4. Such dynamical behaviors physically correspond to the energies of the gravitons in the spectral function, and can be captured by the merging and splitting of resonance peaks in the inelastic photon scattering

**Table 3 | The overlap between the ground states with different filling factors (first row) of the corresponding model Hamiltonian and the Coulomb interaction**

| 2/7(8e) | 4/13(8e) | 2/11(6e) | 2/9(8e) | 1/4(8e) |
|---------|----------|----------|---------|---------|
| 0.953(3a) | 0.986(3b) | 0.957(3c) | 0.993(4a); 0.989(4b) | 0.908(5) |

The corresponding electron numbers and figure indices have been included.

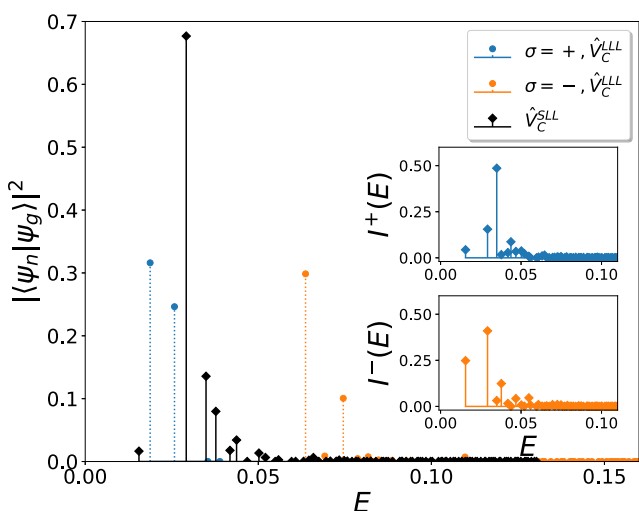

**Fig. 5 | Spectral functions of the Jain state at $\nu = 2/7$ with 8 electrons with respect to the Coulomb interaction in the LLL $\hat{V}_C^{LLL}$ and the second LL (SLL) $\hat{V}_C^{SLL}$.** In the main plot, the chiralities of the two GMs $|\psi_g^\sigma\rangle$ with respect to $\hat{V}_C^{LLL}$ are denoted as $\sigma = +$ (blue) or $-$ (orange). Thus, the GM with respect to $\hat{V}_C^{SLL}$ (black stems in the main figure) can be further resolved by taking the overlap with $|\psi_g^\sigma\rangle$ as the subplots show, from which one can clearly observe that some of the eigenstates can be regarded as two GMs with opposite chiralities at almost the same energy. Source data are provided as a Source Data file.

measurements[22,54–56]. Furthermore, merging and splitting GMs can also be expected for the non-Abelian Pfaffian state at $\nu = 1/4$, which similarly has an exact three-body model Hamiltonian[57–59]. In both cases, the GMs are of the same chirality. For gravitons with opposite chiralities, even if their energies merge, the multiple gravitons can still be distinguished by circularly polarized light[14,15,18]. We will discuss this in more detail next with numerical computations.

While the main concepts and predictions of the GMs have been formulated analytically above, it is also helpful to further illustrate the formalism with examples of numerical calculations. The spectral functions for the Jain states at $\nu = 2/7, 2/9, 1/4$ with Coulomb interaction have been computed[27,28]. Her, we compute the spectral functions of these and additional FQH states with model Hamiltonians on the sphere to show that two or even more peaks can be unambiguously resolved and far separated compared to the realistic interactions. One can expect similar signatures in experiments if the realistic Hamiltonians were adiabatically close to these model Hamiltonians. Otherwise, experimental parameters, such as the sample thickness or the LL mixing, may need to be carefully tuned. In all cases we have studied, the ground states of the model Hamiltonians and the realistic Hamiltonians at the same filling factor have very high overlaps (as shown in Table 3), showing strong evidence that they are in the same topological phases.

The first two examples are the Jain state at $\nu = 2/7$ (the PH conjugate of the Laughlin $\nu = 1/5$ state within $\mathcal{H}_1^{2bdy}$) and the interacting CF state at $\nu = 4/13$ (the PH conjugate of the Laughlin $\nu = 1/7$ state within $\mathcal{H}_1^{2bdy}$, with some experimental evidence[60]). In both cases, the PH conjugate is defined for CFs, each with one electron attached to two

fluxes. The CHS of both phases are proper subspaces of $\mathcal{H}_1^{2bdy}$, but outside of $\mathcal{H}_1^{2bdy}$. Thus, there will be a non-zero component of the graviton outside of $\mathcal{H}_1^{2bdy}$. A short-range two-body interaction with a very dominant $\hat{V}_1^{2bdy}$ can thus easily resolve the two GMs as previously predicted (Fig. 3a, b). The two GMs can also be clearly resolved with $\hat{V}_{LLL}$ since it is dominated by $\hat{V}_1^{2bdy}$.

The Jain state at $\nu = 2/7$ also provides an example to understand gravitons' dynamics with realistic interactions further. Since the PH conjugation reverses the sign of all physical quantities that are odd in time-reversal operations, the two gravitons of this state should have opposite chiralities. We compute the spectral function with respect to the Coulomb interaction in the second LL, where $\hat{V}_1^{2bdy}$ is much less dominant than in the LLL. The overlap between the ground states is around 0.982, with the incompressibility gap remaining robust with 8 electrons. One can clearly see the merging of the energies of the gravitons shown in Fig. 5, which should be highly relevant in experiments. Thus, in the inelastic unpolarized light scattering experiments, we expect to observe two resonance peaks in the LLL but only one resonance peak in the SLL.

With the circularly polarized light, however, one (potentially broadened due to mixing) peak will be observed at almost the same frequency for each polarization in the SLL as shown in the subplots in Fig. 5, where the spectral function of the chiral GMs is given by:

$$I^\sigma(E) = \sum_i |\langle \psi_i^{SLL} | \psi_g^\sigma \rangle|^2 \delta(E - E_i), \quad \sigma = \{+,-\} \quad (12)$$

where $|\psi_g^\sigma\rangle$ denotes the GM with the chirality $\sigma$, i.e., $|\psi_g\rangle = \sum_\sigma c_\sigma |\psi_g^\sigma\rangle$ and the summation with respect to $i$ is over all the eigenstates $|\psi_i^{SLL}\rangle$ of the Coulomb interaction in the SLL at energy $E_i$. $|\psi_g^\sigma\rangle$ can be calculated by looking for the components of $|\psi_g\rangle$ within two well-separated sub-Hilbert spaces $\mathcal{H}^\sigma$ (denoted by the color blue/orange in Fig. 5), and $c_\sigma$ is the corresponding normalization factor. One can thus take $|\psi_g\rangle$ as the superposition of two chiral GMs, and as shown in Fig. 5, $I^\sigma(E)$ gives the resonance amplitude when a GM of the chirality $\sigma$ is excited from the ground state.

The Jain state at $\nu = 2/11$ offers another interesting example: the PH conjugate of the Laughlin $\nu = 1/7$ state within the $\mathcal{H}_3^{2bdy}$, defined for CFs with one electron attached to four fluxes. The two GMs are within and outside of $\mathcal{H}_3^{2bdy}$, so it can be clearly resolved with a model Hamiltonian with a dominant $\hat{V}_3^{2bdy}$ (and a dominant $\hat{V}_1^{2bdy}$ to maintain the ground state gap). However, with $\hat{V}_{LLL}$ the strength of $\hat{V}_3^{2bdy}$ is only slightly larger than that of $\hat{V}_5^{2bdy}$, and it cannot clearly resolve the two GMs (Fig. 3c) using the unpolarized light. This is an example when reducing the $\hat{V}_3^{2bdy}$ of the interaction leads to the mixing and merging of the two GM energies, while the ground state is not affected at all since it is in the null space of $\hat{V}_3^{2bdy}$. Note that the energies of the GMs for this FQH phase are not affected by the $\hat{V}_1^{2bdy}$ component of the two-body interaction. Since the two GMs have opposite chiralities, two resonance peaks at similar energies can still be resolved using the circularly polarized light, as is the case for the $\nu = 2/7$ state with Coulomb interaction in the SLL.

For the Jain state at $\nu = 2/9$, any $\hat{V}_1^{2bdy}$ dominated interaction (e.g., $\hat{V}_{LLL}$) will give two GMs, as can be analytically proven. From numerical studies with eight electrons, the spectral weight of the GM outside $\mathcal{H}_1^{2bdy}$ is small as compared to other Jain states. It is also another example where a single GM (here within $\mathcal{H}_1^{2bdy}$) can be split into two GMs, this time with the introduction of the three-body interactions. We can analytically show that within $\mathcal{H}_1^{2bdy}$, there is non-zero graviton spectral weight both within and outside of $\mathcal{H}_9^{3bdy}$. Thus, an introduction of $\hat{V}_9^{3bdy}$ to the microscopic Hamiltonian can lead to the splitting of the two GMs in total to three GMs, as shown in Fig. 4. It is worth noting that the GM within $\mathcal{H}_9^{3bdy}$ dominates, and both the second and the third GM have spectral weights that are more than one order of magnitude smaller. This could be a finite-size effect since we can analytically show

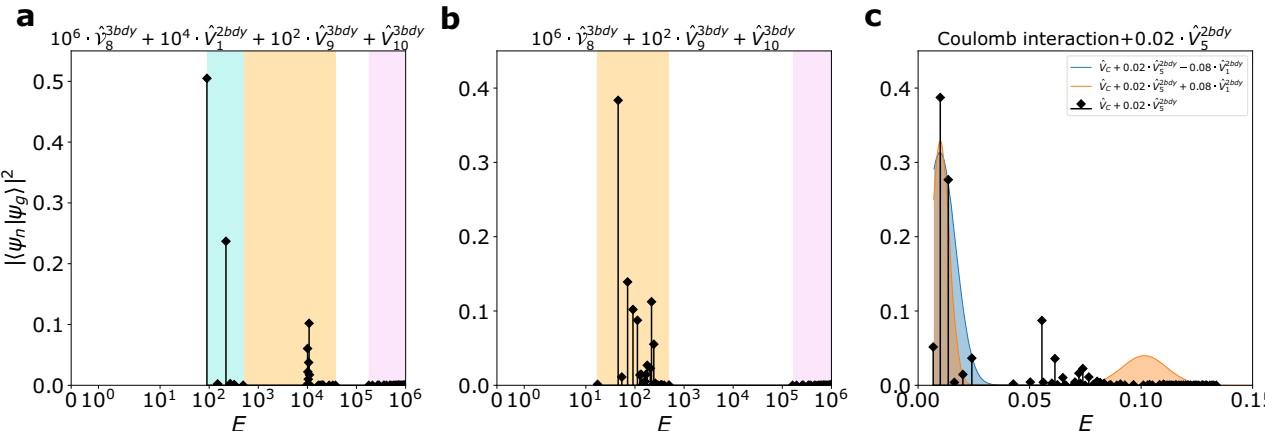

**Fig. 6 | Spectral functions of the Pfaffian state at $\nu = 1/4$ with 8 electrons.**
**a**, **b** Model Hamiltonian with and without $\hat{V}_1^{2bdy}$. The turquoise, the orange and the
pink region denote $\mathcal{H}_1^{2bdy}$, the null space of $\hat{V}_8^{3bdy}$ and its complement corre-
spondingly. When there is no two-body PP $\hat{V}_1^{2bdy}$ in the Hamiltonian, one can clearly
observe the merging of peaks from **a** to **b**. **c** Coulomb interaction ($\hat{V}_C$) with mod-
ifications. Considering $\hat{V}_C$ is $\hat{V}_1^{2bdy}$-dominated, we can also observe two peaks,

which also shows that one can significantly increase (orange area) or decrease (blue
area. Both are smoothed by using Gaussians as in Fig. 4) the separation between the
peaks by properly tuning $\hat{V}_1^{2bdy}$ in experiments. A slightly exaggerated ratio
between different PPs is adopted to clearly show the signature of different CHSs. A
small $\hat{V}_5^{2bdy}$ is added to $\hat{V}_C$ for stabilizing the non-Abelian Pfaffian ground state.
Source data are provided as a Source Data file.

that the spectral weights of all three GMs are non-zero for any finite
systems. It is still possible, however, that in the thermodynamic limit,
the weights of the second and even the third GM vanish. We cannot
numerically access system sizes with more than eight electrons and
will leave more detailed discussions to future works. Moreover, in
experiments, there can be two separated peaks with respect to the
Coulomb interaction with a properly tuned $\hat{V}_1^{2bdy}$ as shown in Fig. 4c,
where we used Gaussian smoothing to model the signal broadening
due to noises in the experimental measurement.

The non-Abelian Pfaffian state at $\nu = 1/4$ is the exact ground state
of $\hat{V}_{10}^{3bdy}$. Just like the Pfaffian state at $\nu = 1/2$ (believed to be stabilized
by second LL Coulomb interaction)[34,61], this state can also be stabilized
by a slightly modified $\hat{V}_{LLL}$, with an overlap of 0.91 for eight electrons.
While this small perturbation may not be easily realized in experi-
ments, with this Hamiltonian, we have the clear understanding that
there will be two GMs (one inside, and the other outside of $\mathcal{H}_1^{2bdy}$). It is
interesting to note that we have the hierarchical relationship that
$\mathcal{H}_9^{3bdy} \subset \mathcal{H}_1^{2bdy} \subset \mathcal{H}_8^{3bdy}$, and both GMs are within $\mathcal{H}_8^{3bdy}$ and at the same
time outside of $\mathcal{H}_9^{3bdy}$. Thus, with the model Hamiltonian consisting of
only three-body PPs, these two GMs will again merge to become a
single GM, in the absence of $\hat{V}_1^{2bdy}$, as reflected in Fig. 6. As a com-
parison, the anti-Pfaffian state at $\nu = 1/4$ (as the particle-hole conjugate
partner of the Pfaffian-1/4 state within $\mathcal{H}_1$) has two peaks with opposite
chiralities. With a slightly modified Coulomb interaction when the
energies of the two gravitons merge, we can see only one reso-
nance peak from the inelastic scattering of the unpolarized light but
one resonant peak each for the two circularly polarized light with the
opposite chirality. Furthermore, properly tuning $\hat{V}_1^{2bdy}$ in realistic
interactions can lead to a better resolution of the peaks, as shown in
Fig. 6c. For non-Abelian states, there are additional neutral modes, e.g.,
the "gravitino" modes at spin $s = 3/2$ for Pfaffian[6,62], that can be con-
sidered as super-partners of the gravitons. We expect multiple GMs
will also lead to multiple gravitino modes and will leave detailed dis-
cussions elsewhere.

## Discussion
The experimental detection of the multiple GMs and their interactions
in FQH systems is particularly interesting because of both the

topological and geometric aspects of such neutral excitations. Inelastic
scattering experiments can be carried out in the FQH state with pho-
nons or photons of proper frequency to check the existence of the
GMs and find their energies. To further detect the chiralities of GMs,
one needs to use circularly polarized light corresponding to the pho-
tons with +2 or −2 spin angular momenta transferred to the
system[14–19,25]. With realistic interactions, different GMs can interact and
mix strongly if their energies are similar, and for GMs of the same
chirality, multiple GMs can merge into one. For GMs of opposite
chiralities, even if their energies are close, they may still be resolved
with circularly polarized light, so the resonance peaks could be broa-
dened due to mixing and scattering between the GMs and the multi-
roton continuum. The microscopic picture we developed points to the
crucial role of the hierarchy of energy scales associated with different
CHSs, which realistic interaction must imitate to resolve multiple GMs.
Thus, the realization of a robust Hall plateau may not be enough to
detect GMs. The experimental system may need to be flexible enough
to tune the effective electron-electron interactions to control the
dynamics of low-lying gapped excitations.

For systems when the LL mixing is negligible (e.g., with a strong
magnetic field), our calculation shows that short-range interaction
(e.g., the Coulomb interaction in the LLL) can at most resolve two GMs
both for abelian and non-Abelian FQH phases, and we have not found
any exceptions. The universality of such results is due to the algebraic
structure of the Laughlin CHSs. To resolve the two GMs clearly, we
prefer to have the effective interaction be as short-range as possible.
Formally if we expand the realistic two-body interaction in the Haldane
PP basis, we should aim to have the ratio consecutive PP coefficients
(i.e., the ratio of the coefficient of $\hat{V}_i^{2bdy}$ over that of $\hat{V}_{i+1}^{2bdy}$) to be large.
This can be achieved by screening electron-electron interaction or by
increasing the sample thickness in experiments. In particular, Jain
states at $\nu = N/(2nN + 1)$ with $N > 1$, as well as PH conjugate states at
$\nu = 2(n − k)/(2(n − k)(2k + 1) + 1)$ with $k > 0$, will all have two GMs.
Numerical calculations also show that short-range interactions favor
the incompressibility of these FQH states compared to the bare Cou-
lomb interaction.

The most easily observed second GM in experiments would be
the one outside of the $\hat{V}_1$ null space (i.e., $\mathcal{H}_1$), which requires a
dominant $\hat{V}_1$ interaction and can be realized with the Coulomb
interaction within the LLL for FQH states around $\nu = 1/4$ as discussed
in previous works[9]. For FQH states in the SLL, however, the GM

outside of $\mathcal{H}_1$ will mix strongly with the ones inside $\mathcal{H}_1$ because of the significantly stronger $\hat{V}_3$ as compared to LLL Coulomb interaction. It is also important to note that while the FQH states around $\nu = 1/6$ (e.g., the $\nu = 2/11$ state), in principle, have at least two GMs (except for the Laughlin state at $\nu = 1/7$), these GMs will be hard to observe even with LLL Coulomb interaction. This is because such GMs have to be resolved by a dominant $\hat{V}_3^{2bdy}$ (as compared to $\hat{V}_{k>3}^{2bdy}$), which is not the case for LLL Coulomb interaction. Thus, in general, observing multiple GMs, even for simple two-body interactions, will require careful tuning of experimental parameters with Coulomb-based interaction.

The short-range two-body interactions do not favor non-Abelian FQH states, as the compressible composite Fermi liquid (CFL) states are generally more competitive, for example, at $\nu = 1/(2n)$[29,63–68]. For non-Abelian FQH states, we generally require longer-range interactions (e.g., Coulomb interaction in the second LL) or few-body interactions from LL mixing[29,35,59]. While it is hard to predict from finite-size numerical calculations how realistic interactions can stabilize these exotic states, these additional ingredients are necessary if we want to observe more than two GMs. A possible candidate for three GMs seems to be the Jain state at $\nu = 2/9$, where we have shown that the proper introduction of three-body interactions can lead to three peaks in the spectral function. However, we expect one of the peaks resolved by the three-body interaction to be relatively weak, and the dominant peak resides at low energies. Our numerical results are inconclusive for this state due to the small system sizes accessible, and more work is needed to establish its behavior from finite-size scaling.

For the effective field theory construction, the number of gravitational fields needed (i.e., the Haldane modes) for a complete description of the response to the metric fluctuation should be determined by the underlying microscopic theory. It is important first to identify the physical Hilbert space on which the effective field theory is based. For example, for the Jain states near $\nu = 1/(2n)$, the elementary particles are CFs with each electron bound to $2n$ magnetic fluxes (and their PH conjugates, or CF holes). The Hilbert space is thus spanned by CF levels (the fully filled ones give the Jain $\nu = N/(2nN+1)$ states) and their PH conjugates (giving Jain states with $\nu = N/(2nN-1)$). We can denote it as the base space of the effective field theory. In particular, it is the full Hilbert space of LLL for $n = 1$. For $n > 1$, the base space is $\mathcal{H}_{n-1}^{2bdy}$.

To determine if and how many Haldane modes need to be added to effective field theory, it all depends on if the long-wavelength limit of the GMP mode lives entirely within the base space. For $n = 1$, this is definitely the case since the GMP mode lives entirely within the LLL, as illustrated in Fig. 7a. In particular, the regularized density operator $\delta\rho_{\mathbf{q}}$ is PH symmetric. For $n > 1$, the GMP modes for all the Laughlin states live entirely within the base space, but $\delta\rho_{\mathbf{q}}$ is no longer PH symmetric within the base space. One can also show rigorously from the microscopic point of view that all Jain states at $\nu = N/(2nN\pm1)$ with $N > 1$ have GMP modes partially outside of the base space, thus requiring at least one additional Haldane mode to be added to the effective field theory as illustrated in Fig. 7b.

In principle, we can have multiple CHSs containing the base space with the hierarchical structure $\mathcal{H}_{base} \subset \mathcal{H}_1 \cdots \subset \mathcal{H}_k$ with $k > 1$, and the GMP mode resides in $\mathcal{H}_k$ having non-zero weights in all $\mathcal{H}_{k' < k}$. We have not found such cases for Jain states with CHSs defined by two-body and three-body PPs, but it may be possible for other FQH states or CHSs defined by few-body PPs involving clusters of more than three electrons. These cases are illustrated by Fig. 7c, where more than one Haldane mode is needed for the effective field theory to agree with microscopic Hamiltonians that can resolve those CHSs in terms of energy.

Numerical calculations are essential in verifying effective field theory predictions. Still, it is important to note that the number of Haldane modes needed for the effective field theory does not necessarily correspond to the number of peaks in the graviton spectral function since the latter depends on the microscopic details of the

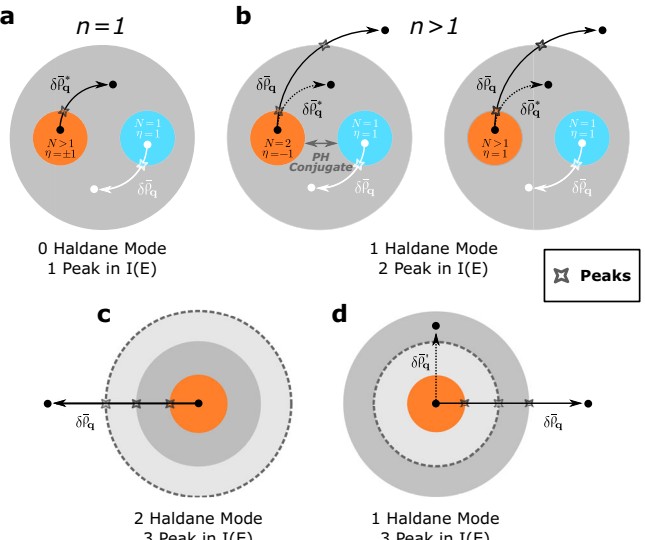

**Fig. 7 | Number of Haldane modes for the Jain states at $N/(2nN+\eta)$ with different structures of CHSs. a** $n = 1$, $N > 1$, $\eta = \pm 1$, e.g., the FQH state at 2/3 and 2/5. The white point denotes the Laughlin states ($N = 1$, $\eta = 1$) in the corresponding CHS (blue circle). Because $\delta\bar{\rho}_{\mathbf{q}} = \delta\bar{\rho}_{\mathbf{q}}^*$, where $\delta\bar{\rho}_{\mathbf{q}}^*$ is the density operator projected to the CHS denoted by the gray circle, these states will show the same behavior as the Laughlin state at 1/3, i.e., one peak with no Haldane mode required. **b** $n > 1$. In this case, a single Haldane mode is needed in the effective field theory despite two peaks observed in the spectral function, among which, given $n$, the states with $N = 2$, $\eta = -1$ can be regarded as the particle-hole conjugate partner of the corresponding Laughlin state within some specific CHS (gray circle), and the states with $N > 1$, $\eta = 1$ are the states in higher CF levels as shown in Fig. 2b. **c**, **d** More possible CHS structures. Thus, the number of Haldane modes added to the effective theory cannot be easily reckoned from the number of peaks in the spectral function $I(E)$, which is instead related to the microscopic Hamiltonian used.

Hamiltonian. Figure 7d is another example that even within the base space, there can be multiple CHSs, which proper model Hamiltonians can resolve. With such interactions, the GM within the base space (the conventional graviton, not the Haldane modes) can lead to multiple spectral weight peaks well-separated in energy. However, the effective theory captures the total weight by coupling the composite particles with the Hall manifold metric.

In fact, from an effective theory point of view, we can always use a single Haldane mode to capture all the GM weights outside of the base space, while the usual composite particle action can capture all the GM weights within the base space. While the known Dirac CF description for the Haldane mode strictly speaking only applies to the FQH states very close to $\nu = 1/(2n)$ (i.e., for Jain states $\nu = N/(2nN\pm1)$ with $N \to \infty$, and does not apply for Laughlin states at $N = 1$), the general arguments here with the relationship between the GMP modes and the base space should apply to all effective field theory description, with or without particle-hole symmetry.

## Data availability

The data of the spectral functions generated in this study are provided in the Supplementary Information as a Source Data file. Source data are provided with this paper.

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

## Acknowledgements

We are grateful to A. Balram and Dung X. Nguyen for fruitful discussions. This work is supported by the Singapore National Research Foundation (NRF) under NRF fellowship award NRF-NRFF12-2020-0005 and a Nanyang Technological University start-up grant (NTU-SUG) (B.Y.).

## Author contributions

Y.W. performed the theoretical and numerical calculations. B.Y. supervised the work.

## Competing interests

The authors declare no competing interests.
