## [Peer Review File · Nature Communications]

REVIEWER COMMENTS

Reviewer #1 (Remarks to the Author):

The manuscript by Wang and Yang investigates graviton modes in fractional quantum Hall fluids. They identify the multiplicity of graviton modes with the different conformal Hilbert spaces within a Landau level. They present both analytic and numerical results to support their ideas. I expect that this work will be of interest to researchers working on the quantum Hall effects.

The work here builds on previous work, quite a bit of which involved at least one of the authors of this work. I felt there were places in the manuscript where the authors could have improved the clarity, and I have tried to identify these in my comments below. One general overall comment - as someone who has worked on the fractional Quantum Hall effect, but not on this particular topic, I found the manuscript a bit opaque at times. I would encourage the authors to consider readers who may be less knowledgeable than themselves in their explanations.

Before I can recommend publication in Nature Communications I think the authors should address the following points.

1) In the first sentence, I think the authors meant to write "remains one of" rather than "remained one of".

2) In the second paragraph, I believe the authors mean the "Newton-Cartan" metric rather than "Newton-Carton"

3) In Fig 2 above each panel, there are abbreviations of the form

"n e m o" where n and m are integers. This abbreviation is never explained, but I presume e is electrons and o is orbitals. The authors should define the abbreviation.

4) In Figs 4 and 6, the authors refer to peaks having been smoothed by Gaussians. I did not understand what the authors meant by this. They should explain what it is they actually did here.

5) The authors use the abbreviation "SLL" but never defined it.

6) It would be helpful if the authors explained how the reader should understand the relationship of their numerical results to their analytic work, especially as regards the multiplicity of graviton modes. At the present time many of the figure captions could do with work to make the results more accessible. For example, in Fig. 4a), the authors state that there are two peaks within $H_1^{\{2\text{bdy}\}}$ with the green region corresponding to the null space of $V_9^{\{3\text{bdy}\}}$ (it isn't explained in this figure what the blue or red regions correspond to). When I look at the plot, I see multiple points corresponding to states at different energies. When the authors refer to a peak, do they mean that there is bunching in the states that have some level of overlap with $|\psi_g\rangle$. If that is the case, should I interpret each "peak" as corresponding to a graviton mode? Based on the way the authors talk about their results, that would seem to make sense, but it would be helpful if they were explicit that this is the case.

7) One thing that also wasn't clear to me. Are the authors claiming that all of the excited states they see in their numerical study are graviton modes? Is it possible to have excited states that aren't graviton modes? If so, how would I distinguish them? Following this line of logic, if I observe excited states of FQH liquids experimentally, how can I be sure I'm measuring graviton modes rather than some other class of modes?

Reviewer #2 (Remarks to the Author):

The existence of multiple magnetorotons in the FQH attracts broad interest for both theorists and experimentalists. For a long time, the GMP magnetoroton was believed to be the unique neutral excitation on FQH. In the recent development of FQH, the new gravitons were proposed in Ref [17] and numerically confirmed in Ref [25] and [26]. Along this line of development, the authors of this paper suggested a microscopic explanation for the new spin-2 modes.

In the paper, the authors analyze the graviton excitations in various fractional quantum Hall (FQH) states using the conformal Hilbert space idea that was introduced in Ref [7] by the same authors. Using the formalism developed in Ref [7], they showed that the number of graviton modes in an FQH state generally depends on the detailed Hamiltonian. Furthermore, using the particle-hole (PH) transformation, they can analytically predict the graviton modes' chirality. I believe that the results in the paper are interesting and timely as well.

However, there are some concerns that I need the authors to clarify.

It seems to me some of the ideas in this paper were published in Ref [7], and the formalism isn't new. So one may ask if this paper is considered novel as standing alone. The author should explain in which way the paper is new and distinct from Ref [7].

In the introduction, the author mentioned the Newton-Cartan metric. Of course, there is a typo that should be the Newton-Cartan metric. Furthermore, the author should replace it with Newton-Cartan formalism since not only was the metric used but a new formalism developed by Son in Ref [8] and was subsequently used by many others.

How do the authors know that their CHS construction is complete? Can one introduce more complicated interactions and split the graviton peaks even further? Then what is the physical meaning of the extra peaks?

The paper showed how to incorporate an emergent metric in 2-body interaction, which gives the meaning of graviton excitations. How does the emergent metric enter the 3-body interaction? Can two different emergent metrics appear at the same time in 4,5-body interactions?

What is the modified Coulomb interaction used in Figs 4(a), 4(b), 6(a), 6(b)? Are they typos?

In Fig 4, how can one determine the graviton peak's chirality appears due to $V^{\{3\text{-bdy}\}}_9$?

In Figures 4(c) and 6(c), what did the data look like before being smoothed by Gaussians? What is the difference between modified Coulomb and Coulomb? I didn't find the explicit definition of modified Coulomb.

In Fig 5, what are the explicit definitions of $I^+(E)$ and $I^-(E)$? How can one numerically obtain $I^+(E)$ and $I^-(E)$? The subplots should present the different chirality peaks for the second Landau level rather than LLL. Furthermore, if one merges $I^+(E)$ and $I^-(E)$, one does not get the black peaks in the main plot. So there should be problems with the definition of $I^+(E)$ and $I^-(E)$ on the second Landau level.

While reading the paper, I also have other concerns. The material presented in this paper would attract the experts in FQH rather than a broad audience. Some detailed arguments of the paper are hard to follow.

In addition to the numerical analysis with Coulomb interaction and the experimental relevance section, the construction of CHS using short-range interactions are quite artificial, with the coefficients that are different by a few orders of magnitude. What are the justifications for that?

Given my above concerns, unfortunately, I would not recommend the paper in its current form for publication in Nature Communications. But I am open mind to hearing the reply from the authors.

Reviewer #3 (Remarks to the Author):

The geometrical aspects of fractional Hall states are particular recent interest. These features have been widely studied using numerical method and this paper provide an alternative microscopic approach to study the neutral excitations of fractional Hall ground state in long wave length limit. The results are well justified and calculations are correct. The structure of the paper is appropriate for its particular community. The authors did a great job exploring their method for multiple different types of fractional Hall state. The authors already explored a wide set of problems using their method.

The only weakness of the paper is regarding its readability for the people who are not in their particular field. I believe the paper structure is very well for its particular experts in the field though. I believe improvement of introduction and/or concluding remarks might help to make the paper more interesting for wider physics audience.

Response to Reviewer #1:

We thank the reviewer very much for his/her time carefully reading our manuscript, and the insightful comments. The referee raised one concern: there is an overlap between this manuscript and one of our previous works. We would like to clarify that our previous work of Ref.[9] (Ref.[7] of the last version. We will always refer to the indices in the revised version below.) mainly introduces an analytical method for computing the variational energies of the single graviton mode. It is a paper on methodology, exploiting the algebraic structure of Laguerre polynomials to derive the variational energy of the SMA modes analytically. Several applications of the methodology in Ref.[9] (the graviton model Hamiltonian, the transition of gapped low-lying excitations) deal with very different topics of multiple graviton modes in this manuscript.

Furthermore, the simple analytic results in Ref.[9] in no way imply the possibility and the microscopic descriptions of the multiple graviton modes. The latter comes from the interplay of the geometric nature of conformal Hilbert spaces, the non-trivial particle-hole symmetry (defined within different CHS, which was not discussed before), and the hierarchical structure defined by the interaction Hamiltonians. We believe all these are new physics that has never been discussed in a unified way to explain multiple graviton modes before.

We also include here a list of new results/concepts proposed in the current manuscript, with no overlap with the previous literature (in particular Ref.[9]):

1. Effective Hamiltonians capturing the dynamical properties of multiple graviton modes;
2. Predictions of the emergence, the merging/splitting behaviour and the chirality of multiple graviton modes for a large family of FQH states;
3. Microscopic basis for the effective field theory and trial wave functions;
4. Experimental relevance on the robustness of multiple graviton modes, etc.

The other general comment from the referee is that manuscript could be a bit opaque at times. Thus in the revised version, we rewrote the introduction part substantially by adding more physical intuitions of the interesting idea of simulating the behaviour of graviton modes in a low-dimensional quantum system, and a guidance paragraph introducing the main goal of each section in the manuscript. The captions of the figures are also polished to make everything self-consistent. More detailed explanations are also added to several places focusing on the technical aspects of this work. We believe that the revised version is now more friendly to a general audience.

In the following we will address all of the reviewer's other points in details:

- **Reviewer's comment:** 1) In the first sentence, I think the authors meant to write "remains one of" rather than "remained one of".

Our response: Thank you very much for pointing it out. We have corrected it in the revised version.

- **Reviewer's comment:** 2) In the second paragraph, I believe the authors mean the "Newton-Cartan" metric rather than "Newton-Carton".

Our response: Yes we are very sorry for the typo. It is corrected in the revised version.

- **Reviewer's comment:** 3) In Fig 2 above each panel, there are abbreviations of the form "n e m o" where n and m are integers. This abbreviation is never explained, but I presume e is electrons and o is orbitals. The authors should define the abbreviation.

Our response: We suppose that the reviewer meant Fig.3 in the manuscript, where we use the notation "n e m o" to denote the system sizes in the numerical calculations, i.e. n electrons and m orbitals (as the reviewer presumes). Indeed the definitions were missing. In the revised version, we added an explanation to this notation in the caption of Fig.3.

- **Reviewer's comment:** 4) In Figs 4 and 6, the authors refer to peaks having been smoothed by Gaussians. I did not understand what the authors meant by this. They should explain what it is they actually did here.

Our response: We thank the referee for pointing this out. We use this Gaussian broadening to make the graph easier to read (since there are many scattered points and sometimes points do overlap). This broadening can also be argued to model the signal broadening by noises in the experimental measurement. It was also used in numerics of finite systems to make the plots look smoother (e.g. see Ref.[28] in the main text)). Thus it is mainly a visualisation tool for the readers without changing the physics of the plot.

The main idea we want to show in Fig.4(c) and 6(c) is that *the modification to the Coulomb interaction (e.g. by adding or subtracting a two-body pseudopotential) can lead to the merging or the splitting of peaks in the spectral function*, which correspond to different graviton modes. This means that as long as the Coulomb interaction can be properly tuned in experiments, it is possible to observe the signature of well-separated graviton modes.

To smoothen the peaks, we use Gaussian functions to fit them, with the mean and the maximum given by the highest stem, and the standard deviation approximately determined by the outmost stem in the same peak. In fact the explicit distribution or shape of the peaks is not important – what matters is the energy range of each peak in the spectral function, because the case that two peaks with minimal overlapping (as the orange parts in Fig.4(c) and 6(c) show) is what we want in experiments to observe double (or multiple) graviton modes.

In the revised version, we have rewritten the caption of these two figures and changed the notations to make them more consistent and less confusing, and also to make sure that readers understand this Gaussian broadening is an ad-hoc visualisation tool.

- **Reviewer's comment:** 5) The authors use the abbreviation "SLL" but never defined it.

Our response: We thank the referee very much for pointing this out. We have revised our manuscript and made sure that all the abbreviations are well defined at their first occurrence, including this word "SLL" (second Landau level).

- **Reviewer's comment:** 6) It would be helpful if the authors explained how the reader should understand the relationship of their numerical results to their analytic work, especially as

regards the multiplicity of graviton modes. At the present time many of the figure captions could do with work to make the results more accessible. For example, in Fig. 4a), the authors state that there are two peaks within $H_1^{\{2\text{bdy}\}}$ with the green region corresponding to the null space of $V_9^{\{3\text{bdy}\}}$ (it isn't explained in this figure what the blue or red regions correspond to). When I look at the plot, I see multiple points corresponding to states at different energies. When the authors refer to a peak, do they mean that there is bunching in the states that have some level of overlap with $|\psi_g\rangle$. If that is the case, should I interpret each "peak" as corresponding to a graviton mode? Based on the way the authors talk about their results, that would seem to make sense, but it would be helpful if they were explicit that this is the case.

Our response: We thank the reviewer for the advice. As the reviewer correctly pointed out, when we refer to a peak in the spectral function, we mean a bunch of states that saturates the overlap with the single-mode approximation (SMA) wave function $|\psi_g\rangle$. This bunch of states are located in one (for a single graviton mode) or more (for multiple graviton modes) narrow energy windows. Note that this is also what we can observe in the scattering experiments when probing the graviton modes. With a properly tuned interaction Hamiltonian (as we used in the numerical calculations) they can be distinguished from each other based on the well separated narrow peaks in the frequency/energy domain. In the revised version, we added explicit definition of a graviton peak in the caption of Fig.3.

The colors in the spectral functions follow the same convention in the manuscript for consistency, so actually the red and the turquoise area in Fig.4 have the same meaning as in Fig.3. But now we understand this might cause unnecessary confusions to the readers, so in the revised version we have added explanations to the colors in all the captions.

- **Reviewer's comment:** 7.1) One thing that also wasn't clear to me. Are the authors claiming that all of the excited states they see in their numerical study are graviton modes? Is it possible to have excited states that aren't graviton modes?

Our response: Yes it is possible. Generally speaking, in an FQH droplet, there exist many types of excitations (See also Ref.[2, 51] in the main text), and the graviton mode is only a specific one in the long-wavelength limit with a spin-2 feature and a geometric origin among the low-lying excitations (i.e. they are hosted by the SMA wave functions which in the long wavelength limit comes from the geometric deformation of the quantum Hall ground state. More details can be found in Ref.[5, 7].).

In our numerical studies, only those eigenstates with substantial overlaps with the single-mode approximation (SMA) wave function (as given in Eq.(7) in the main text) are relevant. As shown in the spectral functions in Fig.3-6, there are many states with very small overlap (likely zero overlap without finite size effect) with the SMA wave function, which are other types of excitations not related to the fluctuation of the quantum metric of the ground state. One example of such states in the $L_z = 2$ sector is the two-roton modes. These are neutral excitations with very small overlap (though not zero for finite systems) with the graviton modes, and with the energy quite far away from the resonance energy of the graviton mode.

- **Reviewer's comment:** 7.2) If so, how would I distinguish them? Following this line of logic, if I observe excited states of FQH liquids experimentally, how can I be sure I'm measuring graviton modes rather than some other class of modes?

Our response: From a physical and numerical point of view, the wave function overlap or the spectral weight is an unambiguous way to distinguish excitations that are graviton modes from those that are not (e.g. those with two rotons). Only excitations of high spectral weight with the ground state metric deformation (i.e. the SMA mode in the long wavelength limit) can be considered as graviton modes. Most excitations of FQH liquids will not have spectral weight with the ground state metric deformation.

There are also proposals to detect the graviton mode in experiments (see Ref.[12-25] in the main text). The goal is to induce quadrupole excitations that are spin-2 and charge neutral in the long wavelength limit (at zero momentum) and try to measure the resonance energy of such induced excitations. There will only be one or a few resonance peaks at specific energies buried in the energy continuum (the continuum consists of states of other types of excitations the reviewer is referring to). The unambiguous experimental signature is to have one or more well separated sharp resonance peaks with long enough lifetime to be detected.

Our work shows that to resolve several resonance peaks (which is actually rather difficult, precisely because of all other excitations the reviewer has pointed out), careful and guided tuning of the interaction (as illustrated in this work) is needed, for example by using sample thickness, screening, Landau level mixing, etc. And the number of graviton modes can be identified as the number of discernable resonance peaks in the corresponding experimental measurement.

We hope the responses above addressed all of the reviewer's concerns, and the revised manuscript is now suitable for publication. If there are any further comments from the reviewer we are happy to discuss with further details. Thank you very much again for your time and consideration.

Response to Reviewer #2:

We appreciate very much the reviewer's careful reading of our manuscript, and the very helpful feedbacks. We also thank the reviewer for finding the paper interesting and timely. There are a number of concerns raised by the reviewer, such as the conformal Hilbert space (CHS) constructions, the emergent metric within CHSs and the chirality of graviton modes, which we will clarify point by point below:

- **Reviewer's comment:** It seems to me some of the ideas in this paper were published in Ref [7], and the formalism isn't new. So one may ask if this paper is considered novel as standing alone. The author should explain in which way the paper is new and distinct from Ref [7].

Our response: We understand the reviewer's concern and would like to clarify that our previous work of Ref.[9] (Ref.[7] of the last version. We will always refer to the indices in the revised version below.) mainly introduces an analytical method for computing the variational energies of the single graviton mode. It is a paper on methodology, exploiting the algebraic structure of Laguerre polynomials to derive the variational energy of the SMA modes analytically. Several applications of the methodology in Ref.[9] (the graviton model Hamiltonian, the transition of gapped low-lying excitations) are used to deal with very different topics of multiple graviton modes in this manuscript.

Furthermore, the simple analytic results in Ref.[9] in no way imply the possibility and the microscopic descriptions of the multiple graviton modes. The latter comes from the interplay of the geometric nature of conformal Hilbert spaces, the non-trivial particle-hole symmetry (defined within different CHS, which was not discussed before), and the hierarchical structure defined by the interaction Hamiltonians. We believe all these are new physics that has never been discussed in a unified way to explain multiple graviton modes before.

We will also include here a list of new results/concepts proposed in the current manuscript, with no overlap with the previous literature (in particular Ref.[9]):

1. Effective Hamiltonians capturing the dynamical properties of multiple graviton modes,
2. Predictions of the emergence, the merging/splitting behaviour and the chirality of multiple graviton modes for a large family of FQH states,
3. Microscopic basis for the effective field theory and trial wave functions,
4. Experimental relevance on the robustness of multiple graviton modes, etc.

We thus believe the two works focus on almost entirely different topics, and some applications of the previously developed theoretical tool in this work for a rather different physical phenomenon should not diminish the seminal contribution of the current work.

- **Reviewer's comment:** In the introduction, the author mentioned the Newton-Carton metric. Of course, there is a typo that should be the Newton-Cartan metric. Furthermore, the author should replace it with Newton-Cartan formalism since not only was the metric used but a new formalism developed by Son in Ref [8] and was subsequently used by many others.

Our response: We thank the reviewer very much for the careful reading of our manuscript and we are sorry for this typo. In the revised version the word has been corrected to “Newton-Cartan formalism”.

- **Reviewer’s comment:** How do the authors know that their CHS construction is complete?

Our response: We thank the reviewer for this nice question. While in principle it is possible to construct a complete hierarchy of CHS, for example by using pseudopotentials involving an arbitrary number of particles (e.g. four-body, five-body pseudopotentials and even more complicated ones), in practice it is rather difficult (and tedious) to check either analytically or numerically. These pseudopotentials are also much less relevant experimentally, so we did not go into details for this manuscript.

As we pointed out in the main text, the dynamical properties of (multiple) graviton modes are completely determined by the Hamiltonian, which in turn determines the relevant CHS with known hierarchical relations. This does not require the knowledge of the complete CHS construction. This is also shown in the comparison of spectral functions of model Hamiltonians and realistic Coulomb interactions in Fig.4-6, where only a small number of CHSs are experimentally relevant.

- **Reviewer’s comment:** Can one introduce more complicated interactions and split the graviton peaks even further? Then what is the physical meaning of the extra peaks?

Our response: In principle this is possible. In this paper we mainly focus on the case of double (e.g. the FQH states at $2/7$, $2/11$, $4/13$ and $1/4$) or triple graviton modes (e.g. the FQH state at $2/9$), but they can be generalised to even more graviton modes (albeit with more artificial interactions). For example, we can add another pseudopotential in the Hamiltonian and create another CHS besides the existing ones. Based on our theory (as shown in Fig.2(a) in the main text) there can be another graviton mode generated in this process. This is natural from the perspective that an independent quantum metric (associated with the additional CHS) is added to the Hamiltonian.

The physical significance of the extra peak is that it corresponds to the excitation created by quantum fluctuation of the metric of the new pseudopotential (or the new CHS). The main message is that each conformal Hilbert space is parametrized by a unimodular metric, and there exist neutral excitations from the quantum fluctuation of such metric (thus defined as graviton modes).

From the experimental perspective, if we create a “gravitational wave” excitation (e.g. by the deformation of the lattice, the anisotropic tuning of the electromagnetic property of the medium/substrate, and the tilt of the magnetic field, etc. Further details can be found in Ref.[14-25] in the main text), and if hypothetically we can experimentally tune the interaction to include additional pseudopotentials defining a new CHS, then there will be observable additional resonance peaks which we can identify as additional gravitons.

- **Reviewer's comment:** The paper showed how to incorporate an emergent metric in 2-body interaction, which gives the meaning of graviton excitations. How does the emergent metric enter the 3-body interaction? Can two different emergent metrics appear at the same time in 4,5-body interactions?

Our response: We thank the reviewer for this question. There are two ways to understand the emergent metrics underlying each of the graviton mode. They are a bit technical (the main ideas can be found in Phys. Rev. B 85, 165318 and Phys. Rev. B 85, 115308), so we did not include too many details in the manuscript (also due to the length limit). Note that the two interpretations introduced below are physically equivalent.

The first (algebraic) way to understand the emergent metrics is by looking into how the conformal Hilbert spaces are defined by the pseudopotentials. The simplest example is the metric of a single Landau level, which was firstly proposed by Haldane (Ref.[7] in the main text). The brief idea is that one can always define a set of ladder operators in such a Hilbert space:

$$\hat{a} = \omega_a^* \hat{R}^a, \quad \hat{a}^\dagger = \omega_a \hat{R}^a \quad (1)$$

where \hat{R}^a denotes the guiding center coordinates and the Einstein's convention has been applied. Note that there exists a complex structure given by $\omega_a \in \mathbb{C}$, **which determines the metric:**

$$g_{ab} = \omega_a^* \omega_b + \omega_a \omega_b^* \quad (2)$$

and also relates to the Levi-Civita symbol by:

$$i\epsilon_{ab} = \omega_a^* \omega_b - \omega_a \omega_b^* \quad (3)$$

The reason why normally this complex structure is ignored is that people tend to presume the rotational invariance or isotropy of the system in the commonly seen textbooks or notes, corresponding to a uniform Euclidean metric. But for a system without such a symmetry, it is natural for a generic metric structure to emerge.

Similarly within a generic CHS, one can construct the conformal algebra of the operators J_{ab} (which are the linear combinations of the product of creation and annihilation operators. Please check Ref.[35] in the main text for more details):

$$[J_{ab}, J_{cd}] = i(\eta_{ad} J_{bc} + \eta_{bc} J_{ad} - \eta_{ac} J_{bd} - \eta_{bd} J_{ac}) \quad (4)$$

where η_{ab} serves as the emergent metric of the CHS, which can be an arbitrary unimodular metric that is different from the Coulomb potential metric or the Galilean metric.

The second (analytic) way is by directly looking into the expressions of the pseudopotentials. We can consider a more general case: to define an n-body interaction projected to a single

Landau level, one needs $q_i = \sqrt{g_i^{ab} q_{i,a} q_{i,b}}$, $i = 1, 2, \dots, n - 1$ (where we have used the Einstein summation), corresponding to the momenta of different particles (or more precisely the momenta of clusters of particles, as the Fourier transform of the Jacobi coordinates). In rotationally invariant systems, the model Hamiltonians are the linear combination of the (generalized) Laguerre polynomials of these momenta, and in different interactions these metrics do not have to be the same. The fluctuations of different metrics can thus lead to different graviton modes. For example, the three-body pseudopotential V_3^{3bdy} (which

punishes the states with the total relative angular momentum equal to 3 among three electrons) can be written as:

$$V_3^{3bdy} = c^{30} \cdot L_3^{(0)}(|\mathbf{q}|^2)L_0^{(0)}(|\mathbf{q}'|^2) e^{-\frac{1}{2}(|\mathbf{q}|^2+|\mathbf{q}'|^2)} + c^{12} \cdot L_1^{(0)}(|\mathbf{q}|^2)L_2^{(0)}(|\mathbf{q}'|^2) e^{-\frac{1}{2}(|\mathbf{q}|^2+|\mathbf{q}'|^2)} \quad (5)$$

where the real coefficients c^{30} and c^{12} can be tuned, $|\mathbf{q}| = \sqrt{g^{ab}q_aq_b}$ (g^{ab} is exactly the metric of this three-body pseudopotential) and $L_m^{(0)}(|\mathbf{q}|^2)$ denotes the Laguerre polynomials.

Similarly when 4-body or 5-body pseudopotential interactions are added to the Hamiltonian, each can have its own unimodular metric that in principle can undergo quantum fluctuations independently.

- **Reviewer's comment:** What is the modified Coulomb interaction used in Figs 4(a), 4(b), 6(a), 6(b)? Are they typos?

Our response: We thank the reviewer very much for his/her carefulness and we are sorry for the confusing legends in Figs 4(a), 4(b), 6(a), 6(b). The modified Coulomb interaction is not used in these four figures, and it is caused by a bug in the plotting program. In the revised version, we have deleted the legends in Fig.4(a), 4(b), 6(a), 6(b).

The modified Coulomb interaction means that the Hamiltonian is the Coulomb interaction with a small two-body pseudopotential correction, and the reason why we need this for some cases (e.g. in Fig. 3(b2) and 6(c)) is that for those filling factors, a pure Coulomb interaction cannot stabilise the FQH phase (at least evidenced from the numerical analysis).

- **Reviewer's comment:** In Fig 4, how can one determine the graviton peak's chirality appears due to $V^{\{3\text{-bdy}\}}_9$?

Our response: We thank the reviewer for raising this question, which exactly shows the usefulness of our microscopic picture. We first would like to emphasise that the prediction of the number of graviton modes based on the microscopic Hamiltonians, as well as the chirality of the gravitons, can be done analytically with our approach. The numerical results in our paper are just validations of our analytical predictions (which are valid in the thermodynamic limit).

By reinterpreting the graviton modes as geometric fluctuation of individual conformal Hilbert spaces, the chirality of a graviton mode can be easily read off from whether or not the ground state comes from a particle-hole conjugate within a certain conformal Hilbert space (CHS), as we tried to explain in the introduction. In this way for any FQH states we can determine the chirality of the graviton modes without the need of any numerical computations. This is because we know that **the chirality of the graviton mode is odd under particle-hole transformation.**

For the FQH state at $2/9$ in Fig.4, it is a state with no PH conjugation in any of the conformal

Hilbert spaces. Thus its multiple graviton modes (the number depends on which Hamiltonian we use as illustrated in Fig.4) will all have the same chirality. In another word whether or not there is a new CHS created by introducing V_9^{3bdy} to the Hamiltonian, the chirality of the graviton modes is always the same. And this chirality is a result of pseudopotential V_9^{3bdy} , it comes entirely from the property of the ground state and is naturally defined for all the graviton modes.

Whether or not multiple gravitons can have different chiralities depends on the ground state. For example, the FQH ground state with $\nu = 2/7$ is the particle-hole conjugate of the Laughlin-1/5 state within the conformal Hilbert space determined by V_1^{2bdy} (but it is not the particle-hole conjugate within the entire lowest LL). Thus the two graviton modes of the FQH state at the filling factor $2/7$ (as shown in Fig.3(a1)) must have opposite chiralities.

To better illustrate the idea of how to determine the graviton peak's chirality, we added more discussions on this topic in the caption of Fig.4 in the revised manuscript.

- **Reviewer's comment:** In Figures 4(c) and 6(c), what did the data look like before being smoothed by Gaussians? What is the difference between modified Coulomb and Coulomb? I didn't find the explicit definition of modified Coulomb.

Our response: The figures before being smoothed by Gaussian are shown below:

Fig.I Spectral function of the FQH states at $\nu = 2/9$ with 8 electrons (the unsmoothed version of Fig. 4(c) in the main text).

Fig.II Spectral function of the Pfaffian states at $\nu = 1/4$ with 8 electrons (the unsmoothed version of Fig. 6(c) in the main text). Here \hat{V}_c denotes the Coulomb interaction, and we abandoned the name “modified Coulomb interaction” by writing down the Hamiltonian explicitly in the figure.

The main idea we want to show in Fig.4(c) and 6(c) is that *the modification to the Coulomb interaction (e.g. by adding or subtracting a two-body pseudopotential) can lead to the merging or the splitting of peaks in the spectral function*, which correspond to different graviton modes, and this means that as long as the Coulomb interaction can be properly tuned in experiments to realise the corresponding gapped topological phases, it is possible to observe the signature of well-separated graviton modes.

As for the second question, the modified Coulomb interactions are the Coulomb interaction plus/minus a small two-body Haldane pseudopotential, which is now properly explained in the revised version.

In the revised manuscript, we have rewritten the caption of these two figures and changed the notation to make them more consistent and less confusing, and also to make sure that readers understand this Gaussian broadening is an ad-hoc visualisation tool. In the main text we also added the explanation to the reason for using the Gaussian broadening.

We will separate the following question into several parts for clarity:

- **Reviewer’s comment:** (1) In Fig 5, what are the explicit definitions of $I^+(E)$ and $I^-(E)$?

Our response: In the last version the definition of the spectral functions is given by:

$$I^\sigma(E) = \sum_{i,n} |\langle \psi_i^{SLL} | \phi_n^\sigma \rangle|^2 \cdot \delta(E - E_i), \quad \sigma \in \{+, -\} \quad (6)$$

where $|\varphi_n^\sigma\rangle$ are the eigenstates with large spectral weights of the gravitons with chirality σ (blue or orange dashed stems in the main plot), and $|\psi_i^{SLL}\rangle$ denotes the eigenstates of the Coulomb interaction in the second Landau level at energy E_i (black stems in the main plot).

In the ideal case where the resonance peak is sharp (i.e. delta function), this definition is equivalent to those in Ref.[16, 25] in the main text (if the quadrupole operators defined in there are properly normalised). In practice, we handpicked the (orange and blue) LLL graviton peaks (which agrees with those obtained from Ref.[16, 25]) with high enough (a bit arbitrary) spectral weights to calculate the spectral function $I^\sigma(E)$, so this definition (while numerically convenient) is not quantitatively rigorous in representing the spectral weights in the inset (the main plots are quantitatively rigorous).

Similar to Ref.[16,25], our old definition gives the qualitative picture of the resonance peaks of gravitons of different chiralities (either well split in the LLL or merged in the SLL), but it was not directly or quantitatively related to experimental measurement.

As the referee is asking about this and also for the paper to be more accessible to a general audience (including the experimentalists), in the revised manuscript we used a different definition that is explicitly and rigorously related to the resonance peaks one could have detected in the experiment, when the gravitons of different chiralities can be resolved (see Ref.[14] and [18]):

$$I^\sigma(E) = \sum_i |\langle \psi_i^{SLL} | \psi_g^\sigma \rangle|^2 \cdot \delta(E - E_i) \quad \sigma \in \{+, -\} \quad (7)$$

where $|\psi_g^\sigma\rangle$ denotes the chiral component of the single mode approximation (SMA) state or the graviton modes with the chirality σ , so that we have the following:

$$|\psi_g\rangle = \sum_\sigma c_\sigma |\psi_g^\sigma\rangle \quad (8)$$

The summation with respect to i is over all the eigenstates $|\psi_i^{SLL}\rangle$ of the Coulomb interaction in the second Landau level at energy E_i . In circularly polarised photon scattering experiments one can excite one of the chiral graviton modes $|\psi_g^\sigma\rangle$ and probe the resonance signal. Such a spectral function explicitly corresponds to the amplitude of the signal that can be measured, because it is proportional to the probability of finding the chiral graviton mode $|\psi_g^\sigma\rangle$ at the state $|\psi_i^{SLL}\rangle$.

In the revised manuscript, we replotted the spectral functions $I^\sigma(E)$ in the subplots of Fig.5 to avoid confusions to the readers. We will explain more about this new definition in the following responses.

- **Reviewer's comment:** (2) How can one numerically obtain $I^{+}(E)$ and $I^{-}(E)$?

Our response: Here we would like to give a detailed introduction to the procedure of plotting Fig.5, which explicitly explains how to numerically get all the spectral functions:

- (iii) Based on the definition of $I(E)$, one can first diagonalize the Coulomb Hamiltonian in the $L_z = 0$ sector in the lowest Landau level to get the ground state $|\psi_0\rangle$ and construct the corresponding single-mode approximation (SMA) wave function

$|\psi_g\rangle = \delta\hat{\rho}_q|\psi_0\rangle$, where $\delta\hat{\rho}_q$ is the regularised density operator. Then by diagonalizing the Coulomb Hamiltonian in the $L_z = 2$ sector one can get the spectrum with all the eigenstates. And there are clearly two graviton peaks, the chirality of which (denoted by orange and blue) can be determined by the conformal Hilbert spaces that they live within.

(ii) The next step is to follow the same routine to diagonalize the Coulomb Hamiltonian in the second Landau level, which gives graviton peaks that are closer to each other, because the pseudopotential V_1^{2bdy} becomes relatively smaller. The corresponding spectral function is shown by the black stems in the main figure.

(iii) Then to determine the spectral weight of the eigenstates of the Coulomb Hamiltonian in the second Landau level and the graviton modes with different chiralities, we calculated the spectral functions $I^\sigma(E)$ of $|\psi_g^\sigma\rangle$. The numerical calculation of such spectral functions is based on the following derivation:

For the SMA state we can separate it into two parts with different chiralities:

$$|\psi_g\rangle = \sum_{\sigma=\pm} c_\sigma |\psi_g^\sigma\rangle \quad (9)$$

The components are orthogonal ($\langle\psi_g^+|\psi_g^-\rangle = 0$) and can be written as:

$$|\psi_g^\sigma\rangle = \frac{1}{c_\sigma} \sum_{j=1}^{N_\sigma} \langle\phi_j^\sigma|\psi_g\rangle \cdot |\phi_j^\sigma\rangle \quad (10)$$

where N_σ is the dimension of the chirality- σ sector (the number of blue/orange stems in the main plot), and $|\phi_j^\sigma\rangle$ are the eigenstates of the Coulomb interaction in the lowest Landau level numerically calculated in the first step, with the coefficients given by:

$$c_\sigma^2 = \sum_{j=1}^{N_\sigma} |\langle\phi_j^\sigma|\psi_g\rangle|^2 \quad (11)$$

Then the spectral function $I^\sigma(E)$ can be written as:

$$I^\sigma(E) = \sum_i |\langle\psi_i^{SLL}|\psi_g^\sigma\rangle|^2 \cdot \delta(E - E_0) = \sum_i \frac{1}{c_\sigma^2} \cdot \left| \sum_{j=1}^{N_\sigma} \langle\phi_j^\sigma|\psi_g\rangle \cdot \langle\psi_i^{SLL}|\phi_j^\sigma\rangle \right|^2 \cdot \delta(E - E_0) \quad (12)$$

and all the overlaps in this expression ($\langle\phi_j^\sigma|\psi_g\rangle$ and $\langle\psi_i^{SLL}|\phi_j^\sigma\rangle$) can be calculated numerically.

In the revised manuscript, we added a paragraph to explicitly show the new definition of $I^\sigma(E)$ with the explanation of its physical significance and experimental relevance.

- **Reviewer's comment:** (3) The subplots should present the different chirality peaks for the second Landau level rather than LLL.

Our response: We are very sorry for the confusion. The subplots were indeed showing the different chirality peaks for the second LL. The caption of Fig.5 in the last version could be misleading, so now we have rewritten it completely. As explained in the last question, because the peaks in the spectral function with respect to the Coulomb interaction in the

lowest Landau level are well separated with opposite chiralities, we would like to show that a relatively smaller V_1^{2bdy} (as in the second Landau level Coulomb interaction) can lead to the mixing of the graviton peaks and also their chiralities. Thus we did use the subplots to present the chiralities of the graviton peaks in the second Landau level rather than the lowest Landau level.

- **Reviewer's comment:** (4) Furthermore, if one merges $I^+(E)$ and $I^-(E)$, one does not get the black peaks in the main plot. So there should be problems with the definition of $I^+(E)$ and $I^-(E)$ on the second Landau level.

Our response: We thank the reviewer very much for this question as this is a subtle point and we should have explained this better in the manuscript.

In the revised manuscript, we strictly require all spectral functions to reflect the resonance measurements in the experiment when a particular state is excited. Thus $I(E)$ is the spectral function when the long wavelength SMA (which in this case is a particular quantum linear combination of two chiral graviton modes, as seen in Eq.(8)) is excited, while $I^\sigma(E)$ is the spectral function when only one of the chiral graviton modes is excited.

In our case, for an eigenstate $|\psi_i^{SLL}\rangle$ of the Coulomb interaction in the second Landau level, the spectral weight of the SMA state (the black stems in the main plot) can be written as (with the delta function omitted):

$$I(E) = |\langle \psi_i^{SLL} | \psi_g \rangle|^2 = |c_+ \cdot \langle \psi_i^{SLL} | \psi_g^+ \rangle + c_- \cdot \langle \psi_i^{SLL} | \psi_g^- \rangle|^2 \\ \neq c_+^2 \cdot \underbrace{|\langle \psi_i^{SLL} | \psi_g^+ \rangle|^2}_{I^+(E)} + c_-^2 \cdot \underbrace{|\langle \psi_i^{SLL} | \psi_g^- \rangle|^2}_{I^-(E)} \neq I^+(E) + I^-(E) \quad (13)$$

So generically speaking, $I^+(E)$ and $I^-(E)$ do not sum up to $I(E)$, which means that if one merges the blue and the orange stems in the subplots, indeed one cannot get the black stems in the main plot. The SMA state is not a classical collection of the chiral graviton modes, but rather a linear superposition of them; and $I^\sigma(E)$ gives the resonance in the energy domain when a specific chiral graviton is excited.

In the revised manuscript, we added the explanation of the relation between $I(E)$ and $I^\sigma(E)$ below Eq.12.

- **Reviewer's comment:** While reading the paper, I also have other concerns. The material presented in this paper would attract the experts in FQH rather than a broad audience. Some detailed arguments of the paper are hard to follow.

Our response: We agree that without some efforts, certain technical details could be difficult to follow, given the rigorous derivations we presented and the novel methods we used. In this revision we tried to provide conceptual, easily accessible arguments integrated with rigorous, technical justifications. Hopefully the interested readers should be able to appreciate the main ideas of this work and could choose to spend more efforts understanding some detailed technicalities if they want.

In the revised version, we rewrote the introduction part substantially by adding more physical intuitions of the interesting idea of simulating the behaviour of graviton modes in a low-dimensional quantum system, and a guidance paragraph helping the readers to choose and understand the main objectives of each section better. Furthermore the captions of the figures are also polished with more details. We also added the discussion to the chiralities of the graviton modes, the introduction to the colors denoting different CHSs, and the systematic explanation of the spectral functions, etc. Following the referee's feedbacks, we have also corrected a number of inconsistent notations and paraphrased a few less precise statements. We believe that the revised version is now more friendly to general audiences.

- **Reviewer's comment:** In addition to the numerical analysis with Coulomb interaction and the experimental relevance section, the construction of CHS using short-range interactions are quite artificial, with the coefficients that are different by a few orders of magnitude. What are the justifications for that?

Our response: The model Hamiltonians we propose here are theoretical tools to show rigorously that interesting graviton dynamics is in principle possible, and that all dynamical aspects of gravitons should depend on the energy scales of different conformal Hilbert spaces (and thus the stiffness of their associated metrics). Thus the construction of CHS (or setting up the coefficients of the pseudopotentials) has to make sure that:

(i) *The ground state is a highest-weight state.*

(ii) *Different conformal Hilbert spaces are well separated in the spectrum to avoid the mixing of graviton modes.*

It is also possible, in principle, that some exotic materials with strong LL mixing can mimic the model Hamiltonians we propose here sufficiently well, and this could be an interesting point to investigate in the future.

We also show that while realistic interaction tends to mix different graviton modes (so they interact with each other), for many cases they can still resolve multiple graviton modes, even if we restrict to Coulomb with or without minor corrections. In Fig.3, Fig.4 and Fig.6 we have included the results with respect to the Coulomb interaction (or with minor corrections) and these results can show very similar signatures to the model Hamiltonians'.

The model Hamiltonians we introduced are in some sense the "best" models for illustrating the physics of multiple graviton modes, and they indeed can be qualitatively identical to the realistic Hamiltonians in certain cases, when the realistic Hamiltonians are "adiabatically" close to the model Hamiltonians. For example, the FQH state at 2/9 is believed to have the model Hamiltonian:

$$H_{2/9} = V_3^{3bdy} + V_5^{3bdy} + V_6^{3bdy} + V_7^{3bdy} + V_8^{3bdy} + V_9^{3bdy} + V_{10}^{3bdy} + V_{11}^{3bdy} \quad (14)$$

(There is no V_4^{3bdy} due to the fermionic statistics.) because one can numerically calculate the overlap between the ground states of the Coulomb interaction and the model Hamiltonian, which can be very high as shown in Table.III in the main text (in this specific case the overlap can be higher than 0.99). The multiple resonance peaks of the gravitons of the two Hamiltonians are also qualitatively similar.

However, our results also show that for certain cases, realistic Hamiltonians will lead to strong graviton-graviton interactions, rendering experimental detections difficult. With the model Hamiltonians we constructed, we can understand and predict such scenario by just looking at the interaction Hamiltonian, without the expensive finite size exact diagonalisation for the entire spectrum (which is infeasible for large system sizes for many cases).

Moreover, in those cases, we propose that more ingenious experimental approaches may be needed (e.g. tuning sample thickness and Landau level mixing) to see multiple graviton modes, and with the model Hamiltonian we know exactly what the targeted tuning should be.

In the revised manuscript, we emphasized these points in more details at the beginning of the “Numerical results and GM interactions” section.

We hope the responses above address all of the referee’s concerns. We do aim to make this paper readable and interesting for a wide range of audience and thank you very much for your time and valuable feedbacks. If there are other places where we can further improve the manuscript, please let us know and we will keep working on it.

Response to Reviewer #3:

We thank the reviewer very much for his/her encouragement. In the revised version, we rewrote the introduction part completely by adding more physical intuitions of the interesting idea of simulating the behaviour of graviton modes in a low-dimensional quantum system, which provides the main purpose of our paper in a more understandable way, and a guidance paragraph introducing the main goal of each section in the manuscript, which could help the readers to choose the sections that they are intrigued by.

Furthermore the captions of the figures are also polished to make everything self-consistent. In particular, we unified the notations in different figures and added the discussion to the chiralities of the graviton modes, the introduction to the colors denoting different CHSs, and the systematic explanation of the spectral functions, etc.

Following the feedbacks of other referees, we also substantially modified certain parts of the paper to make the technical aspects more rigorous and more readable. We believe that the revised version is now accessible to a wide range of audiences.

Thank you very much again for your time and consideration.

REVIEWER COMMENTS

Reviewer #1 (Remarks to the Author):

I am satisfied with the corrections the authors have made and am now happy to recommend the revised manuscript for publication in Nature Communications. My only suggestion is that the English could still be improved in places.

Reviewer #2 (Remarks to the Author):

The authors addressed most of my questions satisfactorily. There is only remaining concern that I have.

In the reply, in the definition of $|\psi_g\rangle$, the authors used the regularised density operator with the subindices q , which I guess is the specific momentum q . To obtain the spectral function, the author may take the limit $q \rightarrow 0$. In this case, the integration of $I^+ I^-$ can be interpreted as the static structure factor at $q=0$, which vanishes as $\sim q^4$ for gapped FQH state as suggested by Ref[5]. Then the definition of I^+ and I^- should be normalized with $1/q^4$ as they did with I . I think it is a minor point.

However, at the moment, I still don't understand how one can obtain $|\psi^\sigma_g\rangle$ and c_σ . Could the author elaborate more on this point?

After this question is addressed properly, I recommend this paper for publication in Nat Comm.

Reviewer #3 (Remarks to the Author):

I reviewed the response of the authors and also the modified manuscript. I believe the authors response is satisfactory. The new version of manuscript is much better written. I recommend the paper for publication in Nature Communications.

Response to Reviewer #2:

Again, we thank the reviewer very much for his/her time carefully reading our revised manuscript. The reviewer has some remaining questions on the technical details, which we will clarify below:

- **Reviewer's comment:** In the reply, in the definition of $|\psi_g\rangle$, the authors used the regularised density operator with the subindices q , which I guess is the specific momentum q . To obtain the spectral function, the author may take the limit $q \rightarrow 0$. In this case, the integration of $I^- + I^+$ can be interpreted as the static structure factor at $q = 0$, which vanishes as $\sim q^4$ for gapped FQH state as suggested by Ref [5]. Then the definition of I^+ and I^- should be normalized with $1/q^4$ as they did with I . I think it is a minor point.

Our response: We thank the reviewer for this question. The reviewer is right about the definition of the GMP (or SMA) wave function $|\psi_g\rangle$, that it is obtained by taking the limit $q \rightarrow 0$. So the normalization factor of $|\psi_g\rangle$ is exactly the static structure factor at $q = 0$, which can be seen from the expression:

$$\lim_{q \rightarrow 0} S_q = \lim_{q \rightarrow 0} \langle \psi_0 | \delta \hat{\rho}_q \delta \hat{\rho}_{-q} | \psi_0 \rangle = \langle \psi_g | \psi_g \rangle.$$

Indeed, we would like to emphasize in the derivation/numerical computation of the spectral functions, only normalized quantum states are used.

From the definition of the spectral functions:

$$\begin{aligned} I(E) &= \sum_n |\langle \psi_n | \psi_g \rangle|^2 \delta(E - E_n) \\ &= \sum_n \langle \psi_g | \psi_n \rangle \cdot \langle \psi_n | \psi_g \rangle \cdot \delta(E - E_n) \\ &= \langle \psi_g | \left[\sum_n |\psi_n\rangle \cdot \langle \psi_n| \cdot \delta(E - E_n) \right] | \psi_g \rangle \end{aligned}$$

where $|\psi_n\rangle$ denotes the eigenstates of the Hamiltonian within a conformal Hilbert space, we can see that the spectral function is given by inserting an operator (which is very similar to a resolution/identity operator without the delta function) into the static structure factor.

As for the normalization issue mentioned by the reviewer, our definition of I^+ and I^- have already considered the normalization problem, which is given by:

$$\begin{aligned} I^\sigma(E) &= \sum_i |\langle \psi_i^{SLL} | \psi_g^\sigma \rangle|^2 \cdot \delta(E - E_0) \\ &= \sum_i \frac{1}{c_\sigma^2} \cdot \left| \sum_{j=1}^{N_\sigma} \langle \phi_j^\sigma | \psi_g \rangle \cdot \langle \psi_i^{SLL} | \phi_j^\sigma \rangle \right|^2 \cdot \delta(E - E_0) \end{aligned}$$

If we understood correctly, the normalization by the coefficients c_σ^2 is exactly what the reviewer asks for. Such details are hidden in the notation $|\psi_i^{SLL}\rangle$ to make the expression look more consistent with $I(E)$. We will explain how to calculate c_σ below. Also, in our numerical results,

all the wave functions are normalized, so the figures in the manuscript should be correct, which can be seen from the fact that the stems in the same set do add up to 1.

- **Reviewer's comment:** However, at the moment, I still don't understand how one can obtain $|\psi_g^\sigma\rangle$ and c_σ . Could the author elaborate more on this point?

Our response: Thank you for the question, we will add a few details in the manuscript to make the point clearer.

To calculate $|\psi_g^\sigma\rangle$ and c_σ , first note that $|\psi_g^+\rangle$ and $|\psi_g^-\rangle$ are two orthogonal components of $|\psi_g\rangle$, each representing a graviton of a particular chirality :

$$|\psi_g\rangle = \sum_{\sigma=\pm} c_\sigma |\psi_g^\sigma\rangle, \quad \langle \psi_g^- | \psi_g^+ \rangle = 0$$

In the "ideal" case (when they have infinite lifetime) $|\psi_g^+\rangle$ and $|\psi_g^-\rangle$ are the eigenstates of the Hamiltonian well separated in energy. In reality they will broaden and each has finite overlap with a small set of eigenstates of the Hamiltonian forming a sub-Hilbert spaces (denoted by \mathcal{H}^σ). Such subspaces are well-defined because they are well separated in energy, and all the eigenstates that span \mathcal{H}^σ (denoted by $|\phi_j^\sigma\rangle$) are accessible from numerics. **We can get these two states $|\psi_g^+\rangle$ and $|\psi_g^-\rangle$ by looking for the component of $|\psi_g\rangle$ within \mathcal{H}^σ , which is apparently given by:**

$$|\psi_g^\sigma\rangle = \frac{1}{c_\sigma} \sum_{j=1}^{N_\sigma} \langle \phi_j^\sigma | \psi_g \rangle \cdot |\phi_j^\sigma\rangle = \frac{1}{c_\sigma} \sum_{j=1}^{N_\sigma} a_j^\sigma \cdot |\phi_j^\sigma\rangle \quad (*)$$

where N_σ is the dimension of \mathcal{H}^σ . In our numerical results shown in Fig.5, $|\phi_j^\sigma\rangle$ are the eigenstates of the Coulomb interaction in the lowest Landau level, given by exact diagonalization and denoted by orange and blue colors. **So the coefficient c_σ is the normalization factor of $|\psi_g^\sigma\rangle$, given by:**

$$c_\sigma^2 = \sum_{j=1}^{N_\sigma} (a_j^\sigma)^2 = \sum_{j=1}^{N_\sigma} |\langle \phi_j^\sigma | \psi_g \rangle|^2 \quad (**)$$

Because all the states in these expressions ($|\phi_j^\sigma\rangle$ and $|\psi_g\rangle$) can be calculated numerically, we can calculate $|\psi_g^\sigma\rangle$ and c_σ by substituting the corresponding overlaps in Eq.(*) and Eq.(**).

In the revised manuscript, we added a more detailed explanation of how to calculate $|\psi_g^\sigma\rangle$ and c_σ below Eq.12.

We hope our responses above clearly addressed the reviewer's concern, and the revised manuscript is now suitable for publication. If there are any further comments from the reviewer, we are happy to discuss with further details. Thank you very much again for your time and consideration.

REVIEWERS' COMMENTS

Reviewer #2 (Remarks to the Author):

I thank the authors for their detailed responses to my questions, and I am satisfied with the revisions made to the manuscript, and I believe it is ready for publication in Nature Communications.